# Rapid Grassmannian Averaging with Chebyshev Polynomials

## Abstract

We propose new algorithms to efficiently average a collection of points on a Grassmannian manifold in both the centralized and decentralized settings. Grassmannian points are used ubiquitously in machine learning, computer vision, and signal processing to represent data through (often low-dimensional) subspaces. While averaging these points is crucial to many tasks (especially in the decentralized setting), existing methods unfortunately remain computationally expensive due to the non-Euclidean geometry of the manifold. Our proposed algorithms, Rapid Grassmannian Averaging (RGrAv) and Decentralized Rapid Grassmannian Averaging (DRGrAv), overcome this challenge by leveraging the spectral structure of the problem to rapidly compute an average using only small matrix multiplications and QR factorizations. We provide a theoretical guarantee of optimality and present numerical experiments which demonstrate that our algorithms outperform state-of-the-art methods in providing high accuracy solutions in minimal time. Additional experiments showcase the versatility of our algorithms to tasks such as $K$-means clustering on video motion data, establishing RGrAv and DRGrAv as powerful tools for generic Grassmannian averaging.

## 1 Introduction

Grassmannian manifolds, which represent sets of $K$-dimensional linear subspaces of $N$-dimensional spaces (Edelman et al., 1998), have been used extensively in machine learning (Huang et al., 2018; Zhang et al., 2018; Slama et al., 2015), computer vision (Harandi et al., 2013; Lui & Beveridge, 2008; Turaga et al., 2011), and signal processing (Gallivan et al., 2003; Mondal et al., 2007; Xu & Hassibi, 2008). Applications include Principal Component Analysis (PCA) (Jolliffe & Cadima, 2016), low-rank matrix completion (Keshavan et al., 2010), multi-task feature learning (Mishra et al., 2019), clustering (Gruber & Theis, 2006), array processing (Love et al., 2003; DeLude et al., 2022), and distance metric learning (Meyer et al., 2009).

An essential primitive operation is finding an average of a collection of points on the manifold. There are several distinct yet reasonable definitions for an average of points on a Grassmannian (Marrinan et al., 2014). Arguably the most natural analog of the Euclidean mean for points on a Riemannian manifold (such as a Grassmannian) is the Fréchet (or Karcher) mean, defined as the point which minimizes the sum of squared distances to all sample points (Fréchet, 1948). Unfortunately, the Fréchet mean rarely admits a closed form solution, instead necessitating approximation via iterative algorithms (Jeuris et al., 2012). Such algorithms are often computationally expensive, scale poorly with dimension, and are not easily decentralized.

The induced arithmetic mean (IAM) is an alternative manifold average computed by first determining the Euclidean mean of the manifold sample points once embedded "naturally" in some Euclidean space and subsequently projecting this Euclidean mean "naturally" back onto the manifold. For Grassmannian manifolds, the standard embedding is the set of projection matrices and the standard projection operation is simply the closest matrix by Frobenius distance (Sarlette & Sepulchre, 2009). This manifold average may be computed much more efficiently in practice and lends itself well to decentralization as the Euclidean mean may be computed by average consensus (Nedic & Ozdaglar, 2009). In such a decentralized setting, computing this Euclidean mean would be the only operation that requires communication in order to compute the IAM.

As the dimensionality of data grows, it becomes increasingly important to consider decentralized algorithms (Nedić et al., 2018) as data might be spread across many machines and only be accessible for processing via distributed algorithms (Beltrán et al., 2023). Distributed computation might be required as well for situations where the data associated to each agent must be treated with privacy protections, where aggregation of all data onto a single node may be prohibited (Han et al., 2017). While a central server is sometimes employed in this regime, it is similarly common for the use of such a server to be infeasible or simply inefficient when compared to fully decentralized approaches (Sun et al., 2021; Feller et al., 2012).

We propose a novel method to efficiently compute the IAM of a collection of points on a Grassmannian manifold. Our method is highly amenable to decentralization, meaning it can be readily deployed to multi-agent systems or used in data centers operating on big data. Our algorithms operate similarly to the famous power method, with the distinction that Chebyshev polynomials are employed to leverage a "dual-banded" property of the problem in order to achieve never-before-seen efficiency in computation and communication. We demonstrate merit through a theoretical guarantee on the optimality of our approach among a class of polynomial-based algorithms, synthetic numerical experiments comparing our algorithms against state-of-the-art, and experiments on real-world problems showcasing the versatility of our algorithms.

## 2 RELATED WORK

The problem of computing an appropriate average on specific manifolds has been investigated for many different manifolds, e.g., spheres $S^N$, special orthogonal matrices $SO(N)$, Stiefel matrices $St(N, K)$, even Grassmannian points $Gr(N, K)$ (Downs, 1972; Buss & Fillmore, 2001; Galperin, 1993; Hueper & Manton, 2004; Absil et al., 2004; Moakher, 2002; Fiori et al., 2014; Yun, 2018; Hauberg et al., 2014). Focus is often given to the Fréchet mean (Chakraborty et al., 2020; Cheng et al., 2016; Le, 2001), however alternatives are becoming increasingly more popular (Fletcher et al., 2008; 2009; Arnaudon et al., 2012; Marrinan et al., 2014; Chakraborty & Vemuri, 2015; Lee & Jung, 2024). Similarly, the problem of consensus on a manifold in a multi-agent setting has been explored in works such as Sepulchre (2011); Tron et al. (2012).

There have been several algorithms proposed for decentralized optimization on manifolds such as Grassmannians. Sarlette & Sepulchre (2009) proposes a decentralized gradient-based algorithm to solve the problem of computing the IAM for connected compact homogeneous manifolds, e.g. $SO(N)$ and $Gr(N, K)$. Deng & Hu (2023) proposes two decentralized gradient-based algorithms for general optimization problems on Riemannian manifolds. Mishra et al. (2019) proposes a decentralized gradient-based gossip algorithm for general optimization problems on a Grassmannian manifold. Similar works include Chen et al. (2021; 2023); Zhang & Sun (2017).

A problem which is closely related to Grassmannian averaging is that of PCA. Ye & Zhang (2021) proposes the DeEPCA algorithm to solve the decentralized PCA problem. While computing a Grassmannian average is not the intended application of DeEPCA, it may be adapted to this task fairly naturally. Gang et al. (2021); Gang & Bajwa (2022); Froelicher et al. (2023) similarly propose distributed algorithms for PCA; for a more comprehensive review of this field, see Wu et al. (2018).

As we will see later in this paper, the problem of Grassmannian averaging is related to the problem of spectral estimation (see Section 3.1). There exist many tailored algorithms in this field, varying based on factors such as eigenvalue vs. eigenvector estimation, matrix rank, symmetry, estimation of leading vs trailing quantities, size of eigengap, etc (Liesen & Strakos, 2013; Lanczos, 1950; Knyazev, 2001; Sleijpen & Van der Vorst, 2000; Zhou & Saad, 2007; Ghanem & Ghosh, 2007; Martinsson & Tropp, 2020). Many such algorithms optimize for centralized computation, using iterative and sometimes stochastic approaches. One of the most elegant solutions in this space is the power method, from which we take inspiration (see Section 3.3).

## 3 BACKGROUND

### 3.1 AVERAGING SUBSPACES

Given a collection of $M$ subspaces, our goal is to determine the average subspace as efficiently as possible. We choose to use the standard IAM definition of "average" (Sarlette & Sepulchre, 2009) as it leads to what we believe is the most efficient algorithm. Formally, let $\mathrm{St}(N, K) := \left\{ \boldsymbol{U} \in \mathbb{R}^{N \times K} \mid \boldsymbol{U}^\mathsf{T}\boldsymbol{U} = \mathbf{I}_K \right\}$ be the set of $N \times K$ Stiefel matrices where $N \geq K$ and let $\mathrm{Gr}(N, K) := \left\{ [\boldsymbol{U}] \mid \boldsymbol{U} \in \mathrm{St}(N, K) \right\}$ (where the equivalence is defined as $[\boldsymbol{U}] := \{\boldsymbol{U}\boldsymbol{Q} \mid \boldsymbol{Q} \in \mathrm{St}(K, K)\}$) be the Grassmannian representing the set of all $K$-dimensional subspaces of $\mathbb{R}^N$. The average of our collection $\{[\boldsymbol{U}_m]\}_{m=1}^M$ is then denoted $[\bar{\boldsymbol{U}}]$ and defined by the following optimization problem

$$[\bar{\boldsymbol{U}}] := \operatorname*{argmin}_{[\boldsymbol{U}] \in \mathrm{Gr}(N,K)} \left\| \left( \frac{1}{M} \sum_{m=1}^M \boldsymbol{U}_m \boldsymbol{U}_m^\mathsf{T} \right) - \boldsymbol{U}\boldsymbol{U}^\mathsf{T} \right\|_{\mathrm{F}}^2 \tag{1}$$

Equation (1) may be manipulated algebraically to be interpreted equivalently in terms of the eigenvectors of $\bar{\boldsymbol{P}} := \frac{1}{M} \sum_{m=1}^M \boldsymbol{U}_m \boldsymbol{U}_m^\mathsf{T}$. Let

$$\bar{\boldsymbol{P}} = \tilde{\boldsymbol{V}} \tilde{\boldsymbol{\Lambda}} \tilde{\boldsymbol{V}}^\mathsf{T} = [\boldsymbol{V} \quad \boldsymbol{V}_\perp] \begin{bmatrix} \boldsymbol{\Lambda} & \mathbf{0} \\ \mathbf{0} & \boldsymbol{\Lambda}_\perp \end{bmatrix} \begin{bmatrix} \boldsymbol{V}^\mathsf{T} \\ \boldsymbol{V}_\perp^\mathsf{T} \end{bmatrix}$$

denote an eigendecomposition where $\boldsymbol{V} \in \mathrm{St}(N, K)$ and the entries of $\tilde{\boldsymbol{\Lambda}}$ are non-increasing. Assuming $\lambda_K > \lambda_{K+1}$ (where $\lambda_k$ denotes the $k$th largest eigenvalue of $\bar{\boldsymbol{P}}$), it can be shown that the solution to eq. (1) is precisely $[\bar{\boldsymbol{U}}] = [\boldsymbol{V}]$ (see Appendix B.2 for a proof). Consequently, determining the span of the leading $K$ eigenvectors of $\bar{\boldsymbol{P}}$ is tantamount to solving eq. (1), which is the perspective we will later use to motivate our algorithms.

As we continue to discuss this problem, it is informative to keep in mind the following properties. The eigenvalues of $\bar{\boldsymbol{P}}$ are conveniently bounded by $\mathbf{0} \preceq \tilde{\boldsymbol{\Lambda}} \preceq \mathbf{I}_N$ and satisfy $\mathrm{tr}\left(\tilde{\boldsymbol{\Lambda}}\right) = K$, which may be determined by inspection. As a result, $\lambda_K, \lambda_{K+1}$ are bounded as $\frac{1}{N-K+1} \leq \lambda_K \leq 1$ and $0 \leq \lambda_{K+1} \leq \frac{K}{K+1}$. For convenience, we occasionally abuse notation to let $[\boldsymbol{X}]$ denote the Grassmannian equivalence class for the span of the columns of arbitrary (not necessarily Stiefel) matrix $\boldsymbol{X} \in \mathbb{R}^{N \times K}$.

### 3.2 AVERAGING SUBSPACES IN A DECENTRALIZED NETWORK

Decentralized or distributed optimization problems arise in numerous real-world scenarios where centralized approaches are impractical or undesirable. These problems are characterized by using information spread across multiple agents or nodes in a network. Motivating reasons include privacy, communication constraints, data storage limitations, scalability, etc.

In the context of this paper, consider the setting where there are $M$ agents, each holding a subspace $[\boldsymbol{U}_m]$, connected by a some undirected communication graph $\mathcal{G}$. We then want each agent to learn the solution $[\bar{\boldsymbol{U}}]$ to eq. (1) under the restriction that each agent only communicate with their neighbors in $\mathcal{G}$.

Average consensus (AC) is a useful primitive in decentralized optimization to quickly approximate the average of real numbers in a decentralized manner (Nedic & Ozdaglar, 2009). If the $m$th agent in a decentralized network holds a matrix $\boldsymbol{A}_m$, AC allows each agent to approximate $\frac{1}{M} \sum_{m=1}^M \boldsymbol{A}_m$ with minimal rounds of neighbor-only communication. Unfortunately, the non-convex manifold structure of $\mathrm{Gr}(N, K)$ precludes us from efficiently applying AC directly to solve eq. (1), as the Euclidean mean of elements from a non-convex set in general does not lay in said set. While we could have each agent compute the matrices $\boldsymbol{U}_m \boldsymbol{U}_m^\mathsf{T}$ and then use AC to approximate $\bar{\boldsymbol{P}}$, this would incur a communication cost of $\mathcal{O}(N^2)$ (the size of $\bar{\boldsymbol{P}}$) which may be much larger than $\mathcal{O}(NK)$. We could instead use AC to average the matrices $\boldsymbol{U}_m$ with preferable communication cost $\mathcal{O}(NK)$, however

the arbitrary choice of representative Stiefel matrix $\boldsymbol{U}_m$ from the Grassmannian equivalence class $[\boldsymbol{U}_m]$ makes this approach ill-posed.

In order to achieve the $\mathcal{O}(NK)$ communication cost without being ill-posed, one can have all agents compute $\boldsymbol{U}_m \boldsymbol{U}_m^\mathsf{T} \boldsymbol{X}$ and then use AC to approximate $\bar{\boldsymbol{P}} \boldsymbol{X}$, where $\boldsymbol{X} \in \mathbb{R}^{N \times K}$ is some matrix agreed upon by all agents a priori. Section 3.3 elaborates on how quantities of the form $\bar{\boldsymbol{P}} \boldsymbol{X}$ can be used to estimate the desired leading eigenvectors. In practice, the requirement that all agents agree upon $\boldsymbol{X}$ might be overly strict; in many scenarios, it often suffices to have each agent $m$ have instance $\boldsymbol{X}_m$ which are all approximately equal (i.e. there exists some $\boldsymbol{X}$ for which $\boldsymbol{X}_m \approx \boldsymbol{X}$ for all $m \in [M]$).

After one iteration of an algorithm, each agent will have some local approximation of the quantity $\bar{\boldsymbol{P}} \boldsymbol{X}$. If the matrix $\boldsymbol{X}$ is retained in memory for each agent, then AC may be applied to a linear combination of the $\bar{\boldsymbol{P}} \boldsymbol{X}$ approximations and $\boldsymbol{X}$ to approximate some quantity $\bar{\boldsymbol{P}}\left(a\bar{\boldsymbol{P}}\boldsymbol{X} + b\boldsymbol{X}\right) = \left(a\bar{\boldsymbol{P}}^2 + b\bar{\boldsymbol{P}}\right)\boldsymbol{X}$. Applying this logic recursively reveals that such an algorithm can approximate $f_t(\bar{\boldsymbol{P}})\boldsymbol{X}$ after $t$ iterations, where $f_t \in \mathcal{P}_t$ is a $t$th order polynomial; in Section 4 we will consider more thoroughly these polynomials and how they can translate to desirable algorithms.

Gradient tracking is a famous technique in decentralized optimization that improves upon the convergence rate of AC-based methods (Shi et al., 2015; Xu et al., 2015; Qu & Li, 2017; Deng & Hu, 2023; Ye & Zhang, 2021). In essence, it sets up a recursion using standard AC whose fixed point satisfies both a consensus condition (meaning all agents agree) and a stationarity condition (meaning the solution is locally optimal). We will later employ a form of gradient tracking over quantities of the form $f_t(\bar{\boldsymbol{P}})\boldsymbol{X}$ for our decentralized algorithms.

### 3.3 THE POWER METHOD

The power method is a classical algorithm to estimate the leading eigenspace of a positive semidefinite matrix $\boldsymbol{A} \in \mathbb{R}^{N \times N}$. A single power iteration applies the matrix $\boldsymbol{A}$ and orthonormalizes (for numerical stability) the result, e.g.

$$\boldsymbol{U}^{(t)} = \mathrm{QR}\Big(\boldsymbol{A}\boldsymbol{U}^{(t-1)}\Big),$$

where $\mathrm{QR}(\cdot)$ computes the $\boldsymbol{Q} \in \mathbb{R}^{N \times K}$ matrix from a QR factorization of the argument. The power method loop may be unrolled (thanks to the property $\mathrm{QR}(\boldsymbol{X}\mathrm{QR}(\boldsymbol{Y})) = \mathrm{QR}(\boldsymbol{X}\boldsymbol{Y})$) to reveal the following form

$$\boldsymbol{U}^{(t)} = \mathrm{QR}\Big(\underbrace{\boldsymbol{A}\boldsymbol{A}\cdots\boldsymbol{A}\boldsymbol{A}}_{t \text{ times}}\boldsymbol{U}^{(0)}\Big) = \mathrm{QR}\Big(\boldsymbol{A}^t\boldsymbol{U}^{(0)}\Big)$$

Let $\boldsymbol{V}_\star \in \mathrm{St}(N, K)$ be a basis for the leading $K$ eigenvectors of $\boldsymbol{A}$. For random initialization $\boldsymbol{U}^{(0)} \in \mathbb{R}^{N \times K}$, the span of $\boldsymbol{U}^{(t)}$ converges to the span of the $\boldsymbol{V}_\star$ provided $\mathrm{rank}\Big(\boldsymbol{V}_\star^\mathsf{T}\boldsymbol{U}^{(0)}\Big) = K$ (Golub & Van Loan, 2013, Chapter 8.2).

At iteration $t$, the power method effectively applies the function $f_t(\lambda) = \lambda^t$ to each eigenvalue of $\boldsymbol{A}$. Consider for example the case where $\lambda_K(\boldsymbol{A}) = 1$ and $\lambda_K(\boldsymbol{A}) - \lambda_{K+1}(\boldsymbol{A}) > 0$. As $t$ increases, the ratio between trailing and leading eigenvalues of $\boldsymbol{A}^t$ shrinks exponentially; formally, for $1 \le k \le K$ and $K + 1 \le \ell \le N$ we have the following

$$\frac{\lambda_\ell(\boldsymbol{A}^t)}{\lambda_k(\boldsymbol{A}^t)} \le \frac{\lambda_{K+1}(\boldsymbol{A}^t)}{\lambda_K(\boldsymbol{A}^t)} = \left(\frac{\lambda_{K+1}(\boldsymbol{A})}{\lambda_K(\boldsymbol{A})}\right)^t < 1$$

While the power method works well, large values of $\lambda_{K+1}$ can slow convergence. We will later show how polynomials other than $\lambda^t$ can overcome this shortcoming and use them in our algorithms.

## 4 METHODS

### 4.1 MOTIVATION

The goal of the RGrAv algorithms is to solve eq. (1) as efficiently as possible. Recall from Section 3 that determining the span of the $K$ leading eigenvectors of $\bar{\boldsymbol{P}}$ is tantamount to solving eq. (1). Mo-

tivated by the decentralized setting and the power method, we restrict our consideration to iterative algorithms which after $t$ iterations compute $f_t(\bar{\boldsymbol{P}})\bar{\boldsymbol{U}}^{(0)}$ for some $t$th-order polynomial $f_t$ and initial estimate $\bar{\boldsymbol{U}}^{(0)}$ (see Section 3.2). If we can choose $f_t$ such that $f_t(\bar{\boldsymbol{P}})$ has its trailing $N - K$ eigenvalues significantly reduced compared to its leading $K$ eigenvalues (relative to initial $\bar{\boldsymbol{P}}$), then the span of the matrix product $f_t(\bar{\boldsymbol{P}})\bar{\boldsymbol{U}}^{(0)}$ will be approximately our desired solution $[\boldsymbol{V}]$ for arbitrary initial $\bar{\boldsymbol{U}}^{(0)}$. We then focus on choosing such a so-called "noise-canceling" polynomial $f_t$, in the sense that the trailing eigenvalues get "canceled".

We consider the case where the spectrum of $\bar{\boldsymbol{P}}$ is dual-banded, i.e. there exists some $0 < \alpha < \beta < 1$ such that $\boldsymbol{\Lambda}_{\perp} \preceq \alpha \boldsymbol{I}_{N-K}$, $\beta \boldsymbol{I}_K \preceq \boldsymbol{\Lambda}$, and $\beta - \alpha \gg 0$ (e.g. $\beta - \alpha = \frac{1}{3}$). These values $\alpha, \beta$ are unknown, but may be estimated heuristically from domain knowledge. This situation can arise, for instance, when points are normally distributed on the manifold (see Section 5.1), where the heuristic estimation of $\alpha, \beta$ comes from estimation of the variance of the dataset. For simplicity, we refer to the intervals $[0, \alpha]$ and $[\beta, 1]$ as the "stop-band" and "pass-band", respectively. Similar to the power method, we would like our polynomial $f_t$ to decrease the ratio between eigenvalues in the stop-band relative to eigenvalues in the pass-band. However, unlike the power method, knowledge of this dual-banded structure (even heuristically) allows us to choose polynomials which optimize the worst case value of this ratio criteria. Leveraging this spectral structure is how we will choose our optimal polynomials $f_t^{\star}$ (see Theorem 1).

There is, however, an important consideration for high-dimensional data. In the case where $N \gg MK$ we are guaranteed that the nullspace of $\bar{\boldsymbol{P}}$ is (at least $N - MK$) high-dimensional by rank subadditivity, meaning there will be a cluster of eigenvalues at 0. For this reason, we will constrain our polynomials $f_t$ to always satisfy $f_t(0) = 0$. Since the ratio criteria of Theorem 1 is invariant to scaling of $f_t$, we provide one final constraint of $f_t(1) = 1$ simply for uniqueness of solution and numerical stability.

**Theorem 1.** For $t \geq 1$, the minimization problem

$$\underset{f_t \in \mathcal{P}_t'}{\text{minimize}} \ \frac{\max_{\lambda \in [0,\alpha]}|f_t(\lambda)|}{\min_{\lambda \in [\beta,1]}|f_t(\lambda)|}, \tag{2}$$

where $\mathcal{P}_t'$ is the set of $t$th order polynomials such that $f_t(0) = 0$ and $f_t(1) = 1$, is solved by

$$f_t^{\star}(\lambda) = \prod_{s=0}^{t-1} \frac{\lambda - r_{s,t}}{1 - r_{s,t}}, \qquad r_{s,t} := \alpha \frac{\cos\left(\frac{\pi(s+1/2)}{t}\right) + \cos\left(\frac{\pi}{2t}\right)}{1 + \cos\left(\frac{\pi}{2t}\right)}$$

which is a modification of a Chebyshev polynomial of the first kind.[1]

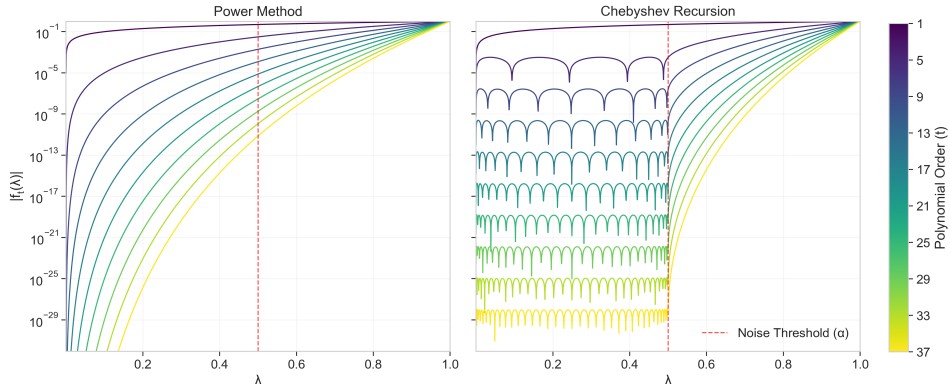

Figure 1: *A visual comparison between the power method and our method. Each method's corresponding $t$th-order polynomial is applied to the eigenvalues $\lambda$ in the domain $[0, 1]$. The Chebyshev recursion with threshold parameter $\alpha = 0.5$ results in the polynomial oscillations being reduced and flattened in the range $[0, \alpha]$.*

---

[1] A proof may be found in Appendix B.1

## 4.2 RGRAV ALGORITHMS

The polynomial $f_t^\star$ may be exactly implemented by iteratively multiplying each factor in the product given in Theorem 1; the "finite" variants of the RGrAv algorithms (Algorithms 5 and 6) do precisely this. While this approach may be acceptable when the number of iterations $t$ is known in advance, intermediate solutions can be quite sub-optimal.

Ideally, one would be able to describe $f_t^\star$ in terms of only a constant number of previous $f_s^\star$, e.g. $f_{t-1}^\star, f_{t-2}^\star$. This would yield an efficient algorithm with optimal intermediate solutions whose memory/compute costs do not grow as $t \to \infty$. Unfortunately, this is not the case; fortunately, empirically, $f_t^\star$ *is well-approximated* in terms of $f_{t-1}^\star, f_{t-2}^\star$ (see Figure 1). For $t \geq 2$, coefficients $a_t, b_t, c_t$ are chosen such that

$$\tilde{f}_t^\star(\lambda) = a_t((\lambda + b_t)f_{t-1}(\lambda) + c_t f_{t-2}(\lambda))$$

matches $f_t^\star(\lambda)$ in its leading three terms, i.e. $f_t^\star(\lambda) - \tilde{f}_t^\star(\lambda) \in \mathcal{P}_{t-3}$ (see Algorithm 7); the "asymptotic" variants of the RGrAv algorithms (Algorithms 1 and 4) use this $\tilde{f}_t^\star$.

As discussed in Section 3.3, orthonormalization must occur periodically for numerical stability. To minimize the frequency of the orthonormalization schedule, our algorithms effectively cache the operation of the most recent exact orthonormalization to a matrix $S$ and then efficiently approximate the orthonormalization procedure for iterations between the schedule by left-application of $S$. We choose the QR factorization for our orthonormalization method, however alternative methods would be acceptable.

In the centralized setting, the RGrAv algorithms need not worry about inaccuracy from average consensus and so the order of operations focuses on minimizing the number of computations performed. In the decentralized setting, the order of operations focuses on minimizing the error in the gradient tracking procedure. Additionally, in the decentralized setting one must take care to use a "stable" orthonormalization method which will not change drastically with small perturbations in the input. We omit the nuances of numerical linear algebra that lead to this problem (see Golub & Van Loan (2013, Chapter 5) for more information) and simply present Algorithm 2, which is a stable wrapper for any implementation of a possibly unstable QR factorization.

---

**Algorithm 1** *Asymptotic DRGrAv (Decentralized Rapid Grassmannian Averaging)*

---

**Input:** $\alpha \in [0, 1)$, $\left\{\bar{U}_m^{(0)}\right\}_{m=1}^M$, $\{U_m\}_{m=1}^M$

**Output:** $\bar{U}^{(T)}$

$\quad S_m \leftarrow \mathbf{I}_K$

$\quad$**for** $t = 1, 2, 3, \ldots$ **do**

$\qquad A_m^{(t)} \leftarrow U_m U_m^\mathsf{T} \bar{U}^{(t-1)}$ $\hfill \triangleright$ Local Power Iteration

$\qquad$**if** $t = 1$ **then**

$\qquad\quad Y_m^{(1)} \leftarrow A_m^{(1)}$ $\hfill \triangleright \left[Y^{(1)}\right] \approx \left[\bar{P}\bar{U}^{(0)}\right]$

$\qquad\quad Z_m^{(1)} \leftarrow Y_m^{(1)}$ $\hfill \triangleright$ Gradient Tracking

$\qquad$**else**

$\qquad\quad a_t, b_t, c_t \leftarrow \text{ChebyshevCoefficients}(t, \alpha)$ $\hfill \triangleright$ See Algorithm 7

$\qquad\quad Y_m^{(t)} \leftarrow a_t\left(A_m^{(t)} + b_t \bar{U}_m^{(t-1)} + c_t \bar{U}_m^{(t-2)}\right)$ $\hfill \triangleright \left[Y^{(t)}\right] \approx \left[\tilde{f}_t^\star(\bar{P})\bar{U}^{(0)}\right]$

$\qquad\quad Z_m^{(t)} \leftarrow \hat{Z}_m^{(t-1)} + Y_m^{(t)} - Y_m^{(t-1)}$ $\hfill \triangleright$ Gradient Tracking

$\qquad$**end if**

$\qquad \hat{Z}_m^{(t)} = \text{AverageConsensus}\left(Z_m^{(t)}\right)$

$\qquad$**if** $t$ is on the orthonormalization schedule **then**

$\qquad\quad \bar{U}_m^{(t)}, S_m \leftarrow \text{StableQR}\left(\hat{Z}_m^{(t)}\right)$ $\hfill \triangleright$ (Numerical Stability)

$\qquad$**else**

$\qquad\quad \bar{U}_m^{(t)} \leftarrow \hat{Z}_m^{(t)} S_m$ $\hfill \triangleright$ (Numerical Stability)

$\qquad$**end if**

$\qquad \bar{U}_m^{(t-1)} \leftarrow \bar{U}_m^{(t-1)} S_m$ $\hfill \triangleright$ (Numerical Stability)

$\quad$**end for**

---

---

**Algorithm 2** *StableQR*

---

**Input:** $\hat{\boldsymbol{Z}} \in \mathbb{R}^{N \times K}$
**Output:** $\boldsymbol{U} \in \mathrm{St}(N, K), \boldsymbol{S} \in \mathbb{R}^{K \times K}$

    $\boldsymbol{Q}, \boldsymbol{R} \leftarrow \mathrm{QR}\left(\hat{\boldsymbol{Z}}\right)$                                      ▷ Arbitrary QR implementation
    $\boldsymbol{D} \leftarrow \mathrm{sgn}(\mathrm{Diag}(\boldsymbol{R}))$
    $\boldsymbol{U} \leftarrow \boldsymbol{Q}\boldsymbol{D}$
    $\boldsymbol{S} \leftarrow \boldsymbol{R}^{-1}\boldsymbol{D}$                                         ▷ Upper triangular inverse

---

## 5 EXPERIMENTS

### 5.1 DECENTRALIZED GRASSMANNIAN AVERAGING

In these experiments, we consider the problem where a network of $M$ connected agents each has a local instance of a Grassmannian basis $\boldsymbol{U}_m \in \mathrm{St}(N, K)$ and we would like for all agents to learn an average Grassmannian basis of all $\boldsymbol{U}_m$ in a strictly decentralized manner (i.e. there is no central server, all communication is neighbor-to-neighbor). Our experiments had parameters $M = 64$, $N = 150$, $K = 30$. To demonstrate the practicality of DRGrAv in both well-connected and sparse communication graphs, we performed experiments for two communication graphs: the hypercube graph and the cycle graph.

We compared DRGrAv to several alternative methods for Grassmannian averaging. Given below are the algorithms, their sources, and considerations for their tuning such that the comparison would be fair.

**DRGrAv** (this paper): Contrary to the following algorithms for which hyperparameters were chosen over large ranges to be empirically optimal, the hyperparameter $\alpha$ is chosen here heuristically as 0.15. We also choose to use the approximate asymptotic variant of DRGrAv, and set the orthonormalization schedule to orthonormalize at every iteration (to match DeEPCA). We believe it is unrealistic in practice to exactly know the optimal choice of $\alpha$, so by comparing our heuristically-tuned algorithm against optimally-tuned competitors (detailed below) we hope to demonstrate that our algorithm is competitive against alternatives, regardless of however optimal their hyperparameter tuning.

**DeEPCA** (Ye & Zhang, 2021): While this algorithm is not intended for decentralized Grassmannian averaging, we found that it could be easily adapted to this setting and gave competitive results: one simply substitutes $\boldsymbol{U}_m\boldsymbol{U}_m^\mathsf{T}$ for the paper's $\boldsymbol{A}_j$. Also, for numerical stability, the paper's QR + SignAdjust procedure is replaced with the StableQR procedure. At a high level, these are all that's needed to adapt the method; see the `deepca.py` for a comprehensive algorithm description.

**DPRGD/DPRGT** (Deng & Hu, 2023): These algorithms are adapted to the decentralized Grassmannian averaging problem by using $-\frac{1}{2}\left\|\boldsymbol{U}_m^\mathsf{T} \bar{\boldsymbol{U}}_m^{(t)}\right\|_\mathrm{F}^2$ for the paper's $f_i$. These algorithms each have a single hyperparameter for step size (referred to as $\alpha$ in Deng & Hu (2023)), which was chosen for each algorithm (up to precision of 2 significant figures) by searching for the value in $\left[10^{-4}, 10^3\right]$ which approximately minimized the MSE; the solutions were on the order of $10^0$.

**COM (Consensus Optimization on Manifolds)** (Sarlette & Sepulchre, 2009): This algorithm is the discrete-time variant of the continuous-time dynamics presented in Equation 20 of Sarlette & Sepulchre (2009). There is a single hyperparameter for step size (referred to as $\alpha$ in Sarlette & Sepulchre (2009)), which was chosen (up to precision of 2 significant figures) by searching for the value in $\left[10^{-4}, 10^3\right]$ which approximately minimized the MSE; the solutions were on the order of $10^{-1}$.

**Gossip** (Mishra et al., 2019): This is Algorithm 1 of Mishra et al. (2019) where $\frac{1}{4}\sum_{m=1}^{M}\sum_{n \in \mathcal{N}(m)} d^2\left(\left[\bar{U}_m^{(t)}\right], \left[\bar{U}_n^{(t)}\right]\right)$ is used as the paper's $g$ (where $\mathcal{N}(m)$ is the set of neighbors of $m$). This algorithm is a gossip algorithm, *not* an average-consensus-based algorithm. As a result, to keep the comparison fair we let each agent perform a gradient step in parallel for each round of consensus the other algorithms perform. In precise terms, during each round of con-

sensus this algorithm will select edges from the graph uniformly at random until there no longer remain any 2 neighboring agents who both have not yet been selected; each of these agents then performs a gradient step and the process repeats. There are 2 hyperparameters[2] for step size $a$ and $b$, which were chosen (up to precision of 2 significant figures) by searching for values in $a \in \left[10^{-4}, 10^{3}\right], b \in \left[10^{-8}, 10^{0}\right]$ which approximately minimized the MSD; the solutions were $a$ on the order of $10^{0}$ and $b$ on the order of $10^{-4}$.

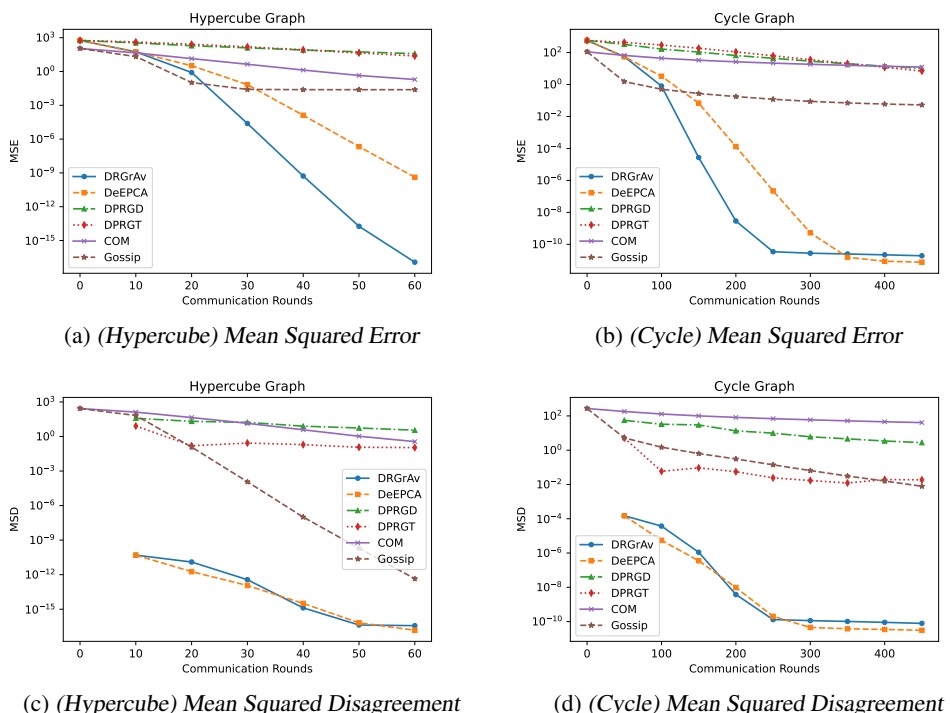

(a) *(Hypercube) Mean Squared Error*  (b) *(Cycle) Mean Squared Error*

(c) *(Hypercube) Mean Squared Disagreement*  (d) *(Cycle) Mean Squared Disagreement*

Figure 2: *Plots of Mean Squared Error/Disagreement for the example decentralized Grassmannian averaging problem. DRGrAv is our proposed algorithm, DeEPCA is from Ye & Zhang (2021), DPRGD and DPRGT are from Deng & Hu (2023), COM is from Sarlette & Sepulchre (2009), and Gossip is from Mishra et al. (2019). The units of the x axes are communication rounds, not algorithm iterations.*

Table 1: *Comparison of runtimes for various algorithms (using the hypercube graph). The first five data columns display time (in milliseconds) until the MSE across agents goes below the given tolerance. COM and Gossip do not in general converge to any specific point, so their metric for tolerance is instead MSD. The minimal quantity in each column is bolded. The final column represents time per algorithm iteration. These decentralized algorithms are not truly run on separate devices, only simulated as such, so these runtimes should be interpreted broadly as general evidence that DRGrAv would perform well in a true decentralized setting.*

| *Time (ms) until tolerance...* | 1e-3 | 1e-6 | 1e-9 | 1e-12 | 1e-15 | *Per Iter.* |
|---|---|---|---|---|---|---|
| **DRGrAv** (This Paper) | **35.4** | 47.4 | **47.4** | **56.7** | **66.1** | *11.8* |
| **DeEPCA** Ye & Zhang (2021) | 35.8 | **45.7** | 53.0 | 61.1 | 80.6 | *9.13* |
| **DPRGD** Deng & Hu (2023) | 1860 | 61200 | >100000 | >100000 | >100000 | *9.24* |
| **DPRGT** Deng & Hu (2023) | 2270 | 2910 | 3470 | 4200 | 4780 | *14.5* |
| **COM*** Sarlette & Sepulchre (2009) | 5050 | 7290 | 9260 | 11000 | 13200 | *16.3* |
| **Gossip*** Mishra et al. (2019) | 1280 | 1730 | 2150 | 2590 | 3390 | *427* |

In order to have a fair comparison, all average-consensus-based algorithms mentioned above used the same consensus protocol. Both graphs used the optimal Laplacian-based communication matrix (i.e. $\boldsymbol{W} = \mathbf{I} - \frac{1}{7}\boldsymbol{L}$ for the hypercube graph, $\boldsymbol{W} \approx \mathbf{I} - \frac{1}{2}\boldsymbol{L}$ for the cycle graph, where $\boldsymbol{L}$ is the

---

[2]Technically, Mishra et al. (2019) has a third hyperparameter, denoted $\rho$. However, their algorithm only ever uses the quantity $\rho a$, so w.l.o.g. we let $\rho = 1$ and control only $a$.

corresponding graph Laplacian matrix) for 10 rounds of communication in the hypercube graph case and 50 rounds of communication in the cycle graph case.

A single synthetic dataset $\{U_m\}_{m=1}^M$ of "normally distributed points with standard deviation $\frac{\pi}{4}$" was used for all experiments. In precise terms, said dataset was generated by sampling a center point $U_C$ uniformly at random on $\text{Gr}(N, K)$ and then computing $U_m := \exp_{U_C}(T_m)$ for all $m \in [M]$, where $T_m := \tilde{U}_m \tilde{\Sigma}_m \tilde{V}_m^\mathsf{T}$ is a random tangent vector at $U_C$ such that $\tilde{U}_m, \tilde{V}_m$ are sampled uniformly at random from sets $\left\{\tilde{U} \mid \tilde{U} \in \text{St}(N, K), U_C^\mathsf{T}\tilde{U} = \mathbf{0}\right\}, \text{St}(K, K)$ respectively and $\tilde{\Sigma} := \text{Diag}\left(\frac{\pi}{4}z\right)$ where $z \sim \mathcal{N}(\mathbf{0}_K, \mathbf{I}_K)$ is a vector draw of $K$ i.i.d. standard normal random variables.

The *Mean Squared Error* quantity at time $t$ was computed as $\frac{1}{M} \sum_{m=1}^M d^2\left(\left[\bar{U}_m^{(t)}\right], \left[\bar{U}\right]\right)$ where $d$ is the extrinsic (or chordal) distance on the Grassmannian defined as $d([U_1], [U_2]) := 2^{-1/2}\left\|U_1 U_1^\mathsf{T} - U_2 U_2^\mathsf{T}\right\|_\text{F}$ and $\bar{U}$ is the true IAM average (see Section 3), computed using the `torch.linalg.eigh` function on $\bar{P}$ directly (runtime of 161 milliseconds). Similarly, the *Mean Squared Disagreement* quantity represents the amount to which the agents' estimates vary at time $t$ and was computed as $\frac{2}{M(M-1)} \sum_{m=1}^M \sum_{n=m+1}^M d^2\left(\left[\bar{U}_m^{(t)}\right], \left[\bar{U}_n^{(t)}\right]\right)$.

The results of these experiments are shown in Figure 2 and Table 1. Six iterations were chosen for the hypercube graph case because at this point DRGrAv reaches floating point tolerance (FPtol), demonstrating that such precision is achievable by an algorithm in such time; Nine iterations were chosen for the cycle graph in order to demonstrate the effective consensus-permitted tolerance (ECPtol) of around $10^{-10}$. DRGrAv performs the best out of all algorithms, converging in the hypercube graph case to FPtol in only 6 iterations and converging in the cycle graph case to ECPtol in only 5 iterations. The adapted DeEPCA method performs most closely to DRGrAv, however still lags behind several orders of magnitude in MSE. In the cycle graph, DeEPCA manages to barely beat DRGrAv in at the end, however given both are more or less at ECPtol we do not think this provides strong evidence to prefer DeEPCA to DRGrAv. Since both COM and Gossip begin with $\bar{U}_m^{(0)} \leftarrow U_m$ instead of some pre-agreed upon starting $U^{(0)}$, they are able to have superior performance to DRGrAv in the short term; however after only 2 iterations this short term behavior ends. DPRGD and DPRGT, being generic algorithms for use on any compact submanifold, do not leverage any of the structure specific to the Grassmannian problem and consequently are not empirically competitive to algorithms which do, e.g. DRGrAv. All algorithms presented have their specific ideal use cases, and we claim that the problem of decentralized Grassmannian averaging is the ideal use case of DRGrAv.

## 5.2 K-MEANS FOR VIDEO MOTION CLUSTERING

We consider the application of the RGrAv algorithm to the problem of video motion analysis, extending the work of Marrinan et al. (2014). Their study applied centralized subspace averaging methods to multiple tasks on the DARPA Mind's Eye video dataset. In our work, we focus specifically on the (centralized) task of $K$-means clustering and compare against the best algorithm presented in their work.

The Mind's Eye dataset consists of a set of "tracklets" — short grayscale videos sequences of moving objects, primarily people. Each tracklet consists of 48 frames of size $32 \times 32$ pixels. To prepare these tracklets for subspace analysis, they are flattened into matrices $X_t \in \mathbb{R}^{1024 \times 48}$, where each column represents a vectorized frame. The subspaces $U_t = \text{span}(X_t)$ spanned by these columns are treated as points on the Grassmannian, effectively encoding the essential motion patterns in the video.

Each tracklet is annotated with a label describing the type of motion it contains, such as "walk" or "ride-bike." These labels provide ground truth for evaluating the effectiveness of clustering algorithms; clusters are considered high-quality if most tracklets in the cluster share the same label. In Marrinan et al. (2014), the authors find that the choice of averaging algorithm does not significantly affect cluster quality as the number of clusters increases, whereas runtime can differ significantly. As a result, we compare to the fastest averaging algorithm from their work – the flag mean – and show that applying RGrAv can reduce runtime for the clustering task.

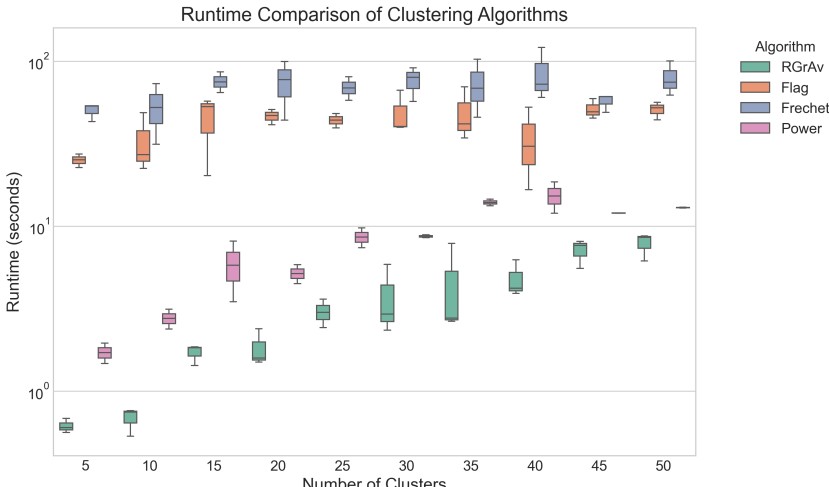

Figure 3: *A comparison of runtime for K-means with various averaging algorithms and numbers of clusters K. The four colors represent the averaging algorithm as RGrAv (green), flag mean (orange), Fréchet mean (blue), and power method (pink). The four algorithms produce clusters with similar quality (excluded for brevity), but the RGrAv algorithm is significantly faster, showing $2\times$-$10\times$ speedup over the other averaging algorithms.*

The standard $K$-means algorithm can be extended to cluster points on the Grassmannian by defining two primitives: a distance metric and an averaging operation. The standard $K$-means algorithm with these operations is shown in Algorithm 3. The centers $\bar{U}_c$ are initialized randomly. At each iteration, the points $U_t$ in the dataset are each assigned to their closest mean using the metric to form clusters. The means are then updated to the average of their respective clusters, and the steps are repeated until the means converge.

---

**Algorithm 3** *Grassmannian K-Means*

---

**Input:** Subspaces $\{U_t\}_{t=1}^T$; Averaging algorithm: $\text{Ave}(U_1, \cdots, U_n)$; Metric: $d(U_1, U_2)$
**Output:** Means $\{\bar{U}_c\}_{c=1}^C$

    $\{\bar{U}_c^{(0)}\}_{c=1}^C \leftarrow \{\text{rand}(\text{St}(N, K))\}_{c=1}^C$            ▷ Initialize
    **while** not converged **do**
        $\{i_t\}_{t=1}^T = \{i : d(U_t, \bar{U}_i^{(k)}) \leq d(U_t, \bar{U}_j^{(k)}) \quad \forall j\}_{t=1}^T$       ▷ Assign clusters
        $\bar{U}_c^{(k+1)} = \text{Ave}(\{U_t : i_t = c\})$           ▷ Compute new means
        converged $= \max_c d(\bar{U}_c^{(k)}, \bar{U}_c^{(k+1)}) < \text{tol}$      ▷ Check termination
        k = k + 1
    **end while**

---

For our clustering experiments, we test on the first 200 tracklets in the Mind's eye dataset, which have a total of 24 unique labels. Given the success of DeEPCA (Ye & Zhang, 2021) in our benchmarks as a runner-up, we use the centralized version (the block power method) in these experiments as well. We test the performance of $K$-means with four different averaging algorithms, namely RGrAv, the power method, the Fréchet mean, and the flag mean. The distance operation for cluster assignment is chosen to be the chordal distance for computational efficiency. The Fréchet mean is computed via iterated gradient descent on the sum of squared distances cost function. Similar to Marrinan et al. (2014), we find that the four averaging algorithms produce clusters with similar quality across various values for $K$ (these results are not shown for brevity). However as can be seen in Figure 3, the runtime varies significantly between algorithms. The Fréchet mean has the slowest runtime regardless of number of clusters, while RGrAv offers a $2\times$-$10\times$ speedup over the other averaging algorithms for all $K$ values.

## 6 Reproducibility Statement

Section 5.1 is intended to be a comprehensive description sufficient for reproducibility; however, in addition all experiments from this section may be reproduced by running the `scripts/decentralized_grassmannian_averaging.py` script in the supplemental code; precise instructions and data formatting are described in the header of this file. The $K$-means experiment can be reproduced in three steps. First is by downloading the SUMMET dataset and putting it in the right subdirectory. Simply follow instructions in `data_sources/video_separation.py`. Next to run the experiment, return to the base directory and run the command `python -m scripts.tracklet_clustering` to run it as a module. Finally once this completes, there should be a `.pkl` file with the results. To create the visualization seen in this paper, run `python -m vis.visualize_tracklet` and look in the new `plots/` subdirectory for the results.

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

## A   ADDITIONAL ALGORITHMS

---

**Algorithm 4** *Asymptotic RGrAv (Rapid Grassmannian Averaging)*

---

**Input:** $\alpha \in [0, 1), \bar{U}^{(0)}, \{U_m\}_{m=1}^M$
**Output:** $\bar{U}^{(T)}$

   $S = \mathbf{I}_K$
   **for** $t = 1, 2, 3, \ldots$ **do**
      $A_m^{(t)} \leftarrow U_m U_m^\mathsf{T} \bar{U}^{(t-1)}$
      $\hat{A}^{(t)} \leftarrow \frac{1}{M} \sum_{m=1}^M A_m^{(t)}$                                            ▷ Power Iteration
      **if** $t = 1$ **then**
         $\hat{Z}^{(1)} \leftarrow \hat{A}^{(1)}$                                ▷ $\left[ \hat{Z}^{(1)} \right] = \left[ \bar{P} \bar{U}^{(0)} \right]$
      **else**
         $a_t, b_t, c_t \leftarrow \text{ChebyshevCoefficients}(t, \alpha)$            ▷ See Algorithm 7
         $\hat{Z}^{(t)} \leftarrow a_t \left( \hat{A}^{(t)} + b_t \bar{U}^{(t-1)} + c_t \bar{U}^{(t-2)} \right)$   ▷ $\left[ \hat{Z}^{(t)} \right] = \left[ \tilde{f}_t^\star(\bar{P}) \bar{U}^{(0)} \right]$
      **end if**
      **if** $t$ is on the orthonormalization schedule **then**
         $\bar{U}^{(t)}, S \leftarrow \text{StableQR}\left( \hat{Z}^{(t)} \right)$                     ▷ (Numerical Stability)
      **else**
         $\bar{U}^{(t)} \leftarrow \hat{Z}^{(t)} S$                               ▷ (Numerical Stability)
      **end if**
      $\bar{U}^{(t-1)} \leftarrow \bar{U}^{(t-1)} S$                               ▷ (Numerical Stability)
   **end for**

---

**Algorithm 5** *Finite RGrAv (Rapid Grassmannian Averaging)*

---

**Inputs:** $\alpha \in [0, 1), T \in \mathbb{N}, \bar{U}^{(0)}, \{U_m\}_{m=1}^M$
**Output:** $\bar{U}^{(T)}$

   $S \leftarrow \mathbf{I}_K$
   **for** $t = 1, 2, 3, \ldots, T$ **do**
      $A_m^{(t)} \leftarrow U_m U_m^\mathsf{T} \bar{U}^{(t-1)}$
      $\hat{A}^{(t)} \leftarrow \frac{1}{M} \sum_{m=1}^M A_m^{(t)}$                                            ▷ Power Iteration
      $r_t \leftarrow \text{ChebyshevRoot}(t, T, \alpha)$                         ▷ See Algorithm 8
      $\hat{Z}^{(t)} \leftarrow \frac{1}{1-r_t} \left( \hat{A}^{(t)} - r_t \bar{U}^{(t-1)} \right)$
      **if** $t$ is on the orthonormalization schedule **then**
         $\bar{U}^{(t)}, S \leftarrow \text{StableQR}\left( \hat{Z}^{(t)} \right)$                    ▷ (Numerical Stability)
      **else**
         $\bar{U}^{(t)} \leftarrow \hat{Z}^{(t)} S$                               ▷ (Numerical Stability)
      **end if**
   **end for**

---

## B   AUXILIARY THEOREMS

**Lemma 2.** Suppose $f^\star \in \mathcal{P}_t'$ is a solution to eq. (2). Then the $f^\star$ has no roots in $[\alpha, 1]$.

*Proof.* If $f^\star$ had a root in $[\beta, 1]$, then $\min_{\lambda \in [\beta, 1]} |f^\star(\lambda)| = 0$ and the objective function is unbounded, so $f^\star$ cannot have any roots in $[\beta, 1]$. This also allows us to conclude $f^\star$ is not the $0$ polynomial, which is used in what follows.

Now, we show that if $f^\star$ has a root in $(\alpha, \beta)$, moving this root to $\alpha$ strictly decreases the

---

**Algorithm 6** *Finite DRGrAv (Decentralized Rapid Grassmannian Averaging)*

---

**Input:** $\alpha \in [0,1)$, $T \in \mathbb{N}$, $\left\{ \bar{U}_m^{(0)} \right\}_{m=1}^{M}$, $\{U_m\}_{m=1}^{M}$

**Output:** $\bar{U}^{(T)}$

   $S_m \leftarrow \mathbf{I}_K$

   **for** $t = 1, 2, 3, \ldots, T$ **do**

      $A_m^{(t)} \leftarrow U_m U_m^{\mathsf{T}} \bar{U}^{(t-1)}$              $\triangleright$ Local Power Iteration

      $r_t \leftarrow \text{ChebyshevRoot}(t, T, \alpha)$              $\triangleright$ See Algorithm 8

      $Y_m^{(t)} \leftarrow \frac{1}{1-r_t} \left( A_m^{(t)} - r_t \bar{U}_m^{(t-1)} \right)$

      **if** $t = 1$ **then**

         $Z_m^{(1)} \leftarrow Y_m^{(1)}$               $\triangleright$ Gradient Tracking

      **else**

         $Z_m^{(t)} \leftarrow \hat{Z}_m^{(t-1)} + Y_m^{(t)} - Y_m^{(t-1)}$      $\triangleright$ Gradient Tracking

      **end if**

      $\hat{Z}_m^{(t)} = \text{AverageConsensus}\left( Z_m^{(t)} \right)$

      **if** $t$ is on the orthonormalization schedule **then**

         $\bar{U}_m^{(t)}, S_m \leftarrow \text{StableQR}\left( \hat{Z}_m^{(t)} \right)$      $\triangleright$ (Numerical Stability)

      **else**

         $\bar{U}_m^{(t)} \leftarrow \hat{Z}_m^{(t)} S_m$            $\triangleright$ (Numerical Stability)

      **end if**

   **end for**

---

**Algorithm 7** *ChebyshevCoefficients*

---

**Input:** $t \geq 2$, $\alpha \in [0,1)$

**Output:** $a_t, b_t, c_t$

   **for** $s = t-2, t-1, t$ **do**

      **if** $s = 0$ **then**

         $g_0 \leftarrow 1$

      **else**

         $r_s \leftarrow \cos\left( \frac{\pi}{2s} \right)$

         $z_s \leftarrow \frac{1+r_s}{\alpha}$

         $\tau_s \leftarrow T_s(z_s - r_s)$      $\triangleright$ $T_s$ is the $s$th-order Chebyshev polynomial of the first kind

         $g_s \leftarrow \frac{\tau_s}{z_s^s}$

      **end if**

   **end for**

   **for** $s = t-1, t$ **do**

      $a_s \leftarrow 2\frac{g_{s-1}}{g_s}$

      $q_s \leftarrow -\frac{r_s}{z_s}$

   **end for**

   $b_t \leftarrow t q_t - (t-1) q_{t-1}$

   **if** $t = 2$ **then**

      $c_t \leftarrow 0$

   **else**

      $c_t \leftarrow \frac{1}{4} a_{t-1} \left( 2t(t-1)(q_t - q_{t-1})^2 - \frac{t}{z_t^2} + \frac{t-1}{z_{t-1}^2} \right)$

   **end if**

---

**Algorithm 8** *ChebyshevRoot*

---

**Input:** $t \in \mathbb{N}$, $T \in \mathbb{N}$, $\alpha \in [0,1)$

**Output:** $r_{t,T}$

   $r_{t,T} \leftarrow \alpha \frac{\cos\left( \frac{\pi(t+1/2)}{T} \right) + \cos\left( \frac{\pi}{2T} \right)}{1 + \cos\left( \frac{\pi}{2T} \right)}$

---

objective function's value showing that $f^\star$ could not have been a solution to the minization problem.

If $f^\star$ had a root in $(\alpha, \beta)$, then we can write $f^\star(\lambda) = (\lambda - \lambda_1)g(\lambda)$ for some $\lambda_1 \in (\alpha, \beta)$ and $g \in \mathcal{P}'_{t-1}$ where $g$ is not the 0 polynomial. It follows that

$$\frac{\max_{\lambda \in [0,\alpha]}|f^\star(\lambda)|}{\min_{\lambda \in [\beta,1]}|f^\star(\lambda)|} = \frac{\max_{\lambda \in [0,\alpha]}|(\lambda - \lambda_1)g(\lambda)|}{\min_{\lambda \in [\beta,1]}|(\lambda - \lambda_1)g(\lambda)|}$$

$$= \frac{\sup_{\lambda \in [0,\alpha)}|(\lambda - \alpha)g(\lambda)|\left|\frac{\lambda - \lambda_1}{\lambda - \alpha}\right|}{\min_{\lambda \in [\beta,1]}|(\lambda - \alpha)g(\lambda)|\left|\frac{\lambda - \lambda_1}{\lambda - \alpha}\right|}$$

$$> \frac{\max_{\lambda \in [0,\alpha]}|(\lambda - \alpha)g(\lambda)|}{\min_{\lambda \in [\beta,1]}|(\lambda - \alpha)g(\lambda)|}$$

where the final line results from the fact that $\left|\frac{\lambda - \lambda_1}{\lambda - \alpha}\right| < 1$ for $\lambda \in [\beta, 1]$ and $\left|\frac{\lambda - \lambda_1}{\lambda - \alpha}\right| > 1$ for $\lambda \in [0, \alpha)$. Now we observe that the function $h(\lambda) := (\lambda - \alpha)g(\lambda) \in \mathcal{P}'_t$ achieves a strictly lower value for the objective function via moving the root $\lambda_1 \in (\alpha, \beta)$ to $\lambda = \alpha$. Thus, $f^\star$ would not be a solution to eq. (2) and therefore $f^\star$ cannot have roots in $(\alpha, \beta)$.

Now we show that if $f^\star$ had a root at $\lambda = \alpha$, we could slightly move the root to some point to the left of $\alpha$ and decrease the objective function's value.

Suppose $f^\star$ has a root at $\lambda = \alpha$. We can write $f^\star(\lambda) = (\lambda - \alpha)g(\lambda)$ for some $g \in \mathcal{P}'_{t-1}$, where $g$ is not the 0 polynomial. Let $\delta > 0$ be such that

$$\max_{\lambda \in [\alpha - \frac{\delta}{2}, \alpha]}|f^\star(\lambda)| < \min_{\epsilon \in [0,\alpha]} \max_{\lambda \in [0,\alpha]}|(\lambda - \epsilon)g(\lambda)|$$

which exists since $\min_{\epsilon \in [0,\alpha]} \max_{\lambda \in [0,\alpha]}|(\lambda - \epsilon)g(\lambda)| > 0$ (as $g \not\equiv 0$ from the beginning of the proof) and $f^\star$ is continuous. It follows that

$$\frac{\max_{\lambda \in [0,\alpha]}|f^\star(\lambda)|}{\min_{\lambda \in [\beta,1]}|f^\star(\lambda)|} = \frac{\max_{\lambda \in [0,\alpha]}|(\lambda - \alpha)g(\lambda)|}{\min_{\lambda \in [\beta,1]}|(\lambda - \alpha)g(\lambda)|}$$

$$= \frac{\sup_{\lambda \in [0,\alpha] \setminus \{\alpha - \delta\}}|(\lambda - (\alpha - \delta))g(\lambda)|\left|\frac{\lambda - \alpha}{\lambda - (\alpha - \delta)}\right|}{\min_{\lambda \in [\beta,1]}|(\lambda - (\alpha - \delta))g(\lambda)|\left|\frac{\lambda - \alpha}{\lambda - (\alpha - \delta)}\right|}$$

$$= \frac{\max\{\sup_{\lambda \in [0,\alpha - \frac{\delta}{2}) \setminus \{\alpha - \delta\}}|(\lambda - (\alpha - \delta))g(\lambda)|\left|\frac{\lambda - \alpha}{\lambda - (\alpha - \delta)}\right|, \Phi\}}{\min_{\lambda \in [\beta,1]}|(\lambda - (\alpha - \delta))g(\lambda)|\left|\frac{\lambda - \alpha}{\lambda - (\alpha - \delta)}\right|}$$

$$> \frac{\max_{\lambda \in [0,\alpha]}|(\lambda - (\alpha - \delta))g(\lambda)|}{\min_{\lambda \in [\beta,1]}|(\lambda - (\alpha - \delta))g(\lambda)|}$$

with $\Phi = \max_{\lambda \in [\alpha - \frac{\delta}{2}, \alpha]}|(\lambda - (\alpha - \delta))g(\lambda)|\left|\frac{\lambda - \alpha}{\lambda - (\alpha - \delta)}\right|$. The final line results from the fact that $\left|\frac{\lambda - \alpha}{\lambda - (\alpha - \delta)}\right| < 1$ for $\lambda \in [\beta, 1]$, $\left|\frac{\lambda - \alpha}{\lambda - (\alpha - \delta)}\right| \geq 1$ for $\lambda \in [0, \alpha - \frac{\delta}{2})$, and since $\left|\frac{\lambda - \alpha}{\lambda - (\alpha - \delta)}\right| \leq 1$ for $\lambda \in [\alpha - \frac{\delta}{2}, \alpha]$,

$$\max_{\lambda \in [\alpha - \frac{\delta}{2}, \alpha]}|(\lambda - (\alpha - \delta))g(\lambda)|\left|\frac{\lambda - \alpha}{\lambda - (\alpha - \delta)}\right| < \min_{\epsilon \in [0,\alpha]} \max_{\lambda \in [0,\alpha]}|(\lambda - \epsilon)g(\lambda)| * 1$$

$$\leq \max_{\lambda \in [0,\alpha]}|(\lambda - (\alpha - \delta))g(\lambda)|$$

Now we observe that the function $h(\lambda) := (\lambda - (\alpha - \delta))g(\lambda) \in \mathcal{P}'_t$ achieves a strictly lower value for the objective function via moving the root from $\alpha$ to $\lambda = \alpha - \delta$. Thus, $f^\star$ would not be a solution to eq. (2) and therefore $f^\star$ cannot have a root at $\lambda = \alpha$. $\qquad \square$

**Definition 1.** A polynomial $f$ is $t$-**equioscillatory** on an interval $[a, b]$ if there exists some $a \le \gamma_0 < \gamma_1 < \cdots < \gamma_{t-1} \le b$ such that $|f(\gamma_s)| = \max_{\lambda \in [a,b]} |f(\lambda)|$ for all $0 \le s \le t - 1$ and $f(\gamma_0) = -f(\gamma_1) = f(\gamma_2) = -f(\gamma_3) = \dots$.

**Lemma 3.** A solution to the minimization problem

$$\underset{f_t \in \mathcal{P}'_t}{\text{minimize}} \frac{\max_{\lambda \in [0, \alpha]} |f_t(\lambda)|}{\min_{\lambda \in [\beta, 1]} |f_t(\lambda)|}$$

must be $t$-equioscillatory.

*Proof.* Suppose $f^\star$ is a minimizer in to eq. (2) that is not $t$-equioscillatory. First, we assume that without loss of generality, $f^\star(\alpha) > 0$ since if $f^\star(\alpha) < 0$, we could replace $f^\star$ with $-f^\star$ and it would still be a minimizer to the eq. (2), and we cannot have $f^\star(\alpha) = 0$ by Lemma 2.

Let $\{\lambda_1 = 0, \lambda_2, \cdots, \lambda_m\} \in [0, \alpha]$ for some $m \le t$ denote the distinct roots of $f^\star$ in increasing order that lie in $[0, \alpha]$. We have $m$-intervals, $\mathcal{I} = \{I_i\}_{i=1}^m$ where $I_i = [\lambda_i, \lambda_{i+1}]$ and $\lambda_{m+1} = \alpha$. We have that for any $i \in [m]$, $\text{sgn}(f^\star(\lambda)) = c$ for every $\lambda \in \text{int}(I_i)$ where $c \in \{-1, 1\}$. In other words, in the interior of any intervals from $\mathcal{I}$, $f^\star$ takes strictly all positive values or strictly all negative values. Define, for any interval $I \subseteq \mathbb{R}$ and function $g$

$$\text{sgn}(g|_I) = c$$

where $c \in \{-1, 1\}$ is the value such that $\text{sgn}(f^\star(\lambda)) = c$ for every $\lambda \in \text{int}(I)$. If such a value does not exist, $\text{sgn}(g|_I)$ is left undefined.

We also define for any compact $A \subseteq \mathbb{R}$ and $g \in C(\mathbb{R})$,

$$\|g\|_A := \max_{x \in A} |g(x)|$$

Let

$$L = \{I \in \mathcal{I} : \max_{\lambda \in I} |f^\star(\lambda)| < \|f^\star\|_{[0, \alpha]}\}$$

$$M = \{I \in \mathcal{I} : \max_{\lambda \in I} |f^\star(\lambda)| = \|f^\star\|_{[0, \alpha]}\} = \mathcal{I} \setminus L$$

$$J = \bigcup_{I \in L} I$$

Define $g(\lambda) = |f^\star(\lambda)| - \|f^\star\|_{[0, \alpha]}$ and denote $\epsilon = \min_{\lambda \in J} |g(\lambda)|$, i.e the minimum distance by which any point in $J$ misses one of $\pm \|f^\star\|_{[0, \alpha]}$.

Let $k = |\{\lambda \in [0, \alpha] : f^\star(\lambda) = \|f\|_{[0, \alpha]}\}|$. Note that $k < t$ by assumption. Let $M = \{M_1, \cdots, M_k\}$ be intervals listed from left to right where $i_j \in [m]$ are indices such that $M_j = I_{i_j} = [\lambda_{i_j}, \lambda_{i_j+1}]$ for $j \in [k]$ and $\lambda_{i_j} < \lambda_{i_m}$ for $1 \le j < m \le k$.

Now define

$$R = \{\lambda_{i_j} : \text{sgn}(f^\star|_{M_j}) \ne \text{sgn}(f^\star|_{M_{j-1}}) \text{ for some } 2 \le j \le k\}$$

We have that since $k \le t - 1$ by assumption, then $|R| \le k - 1 \le t - 2$.

With $R = \{r_1, \cdots, r_q\}$ for some $q \le t - 2$, we define a polynomial $r : \mathbb{R} \to \mathbb{R}$ based on the sign of $f^\star$ on $M_k$.

Case A: If $\text{sgn}(f^\star|_{M_k}) = 1$, we define

$$r(\lambda) := c_r \lambda(\lambda - \beta) \Pi_{i=1}^q (\lambda - r_i)$$

Case B: Otherwise (when $\text{sgn}(f^\star|_{M_k}) = -1$), define

$$r(\lambda) := c_r \lambda(\lambda - 1) \Pi_{i=1}^q (\lambda - r_i)$$

In either case, $r$ be a polynomial of degree at most $t$. We now select $c_r \in \mathbb{R}$ so that $\|r(\lambda)\|_{[0, \alpha]} < \epsilon$ and set the $\text{sgn}(c_r)$ so that $\text{sgn}(r|_{M_1}) = -\text{sgn}(f^\star|_{M_1})$. We now show $\text{sgn}(r|_{M_j}) = -\text{sgn}(f^\star|_{M_j})$ for every $j \in [k]$ via induction. Our base case holds via how we set $c_r$ above.

Suppose inductively that $\text{sgn}(r|_{M_{j-1}}) = -\text{sgn}(f^\star|_{M_{j-1}})$ for some $j \le k$. Then, if $\text{sgn}(f^\star|_{M_j}) = \text{sgn}(f^\star|_{M_{j-1}})$, we have that as no root was added to $r$ between these intervals and so

$$\text{sgn}(r|_{M_j}) = \text{sgn}(r|_{M_{j-1}}) = -\text{sgn}(f^\star|_{M_{j-1}})$$

with the last equality by the induction hypothesis. Otherwise, if $\mathrm{sgn}\big(f^\star|_{M_j}\big) \neq \mathrm{sgn}\big(f^\star|_{M_{j-1}}\big)$, then since a root is added between them

$$\mathrm{sgn}\big(r|_{M_j}\big) = -\mathrm{sgn}\big(r|_{M_{j-1}}\big) = \mathrm{sgn}\big(f^\star|_{M_{j-1}}\big) = -\mathrm{sgn}\big(f^\star|_{M_j}\big)$$

So in both cases, we have $\mathrm{sgn}\big(r|_{M_j}\big) = -\mathrm{sgn}\big(f^\star|_{M_j}\big)$ which closes the induction.

Now we obtain that for any $j \in [k]$,

$$\|f^\star + r\|_{M_j} < \|f^\star\|_{[0,\alpha]}$$

as, for $M_j \neq [\lambda_m, \alpha]$, the maximum $\|f^\star\|_{[0,\alpha]}$ is achieved by $|f^\star|$ in the interior of $M_j$ where $|r| > 0$ as $r$ has no roots in this interior and $\mathrm{sgn}\big(r|_{M_j}\big) = -\mathrm{sgn}\big(f^\star|_{M_j}\big)$. Additionally, one recognizes that if $M_j = [\lambda_m, \alpha]$, then $r(\lambda) < 0$ for $x \in (\lambda_m, \alpha]$ and so the above bound still holds even when the maximum is attained on the boundary of the interval, i.e when $f^\star(\alpha) = \|f^\star\|_{[0,\alpha]}$. Furthermore, we have that

$$\|f^\star + r\|_J \leq \max_{\lambda \in J}|f^\star(\lambda)| + \|r\|_{[0,\alpha]}$$

$$< \max_{\lambda \in J}|f^\star(\lambda)| + \epsilon = \|f^\star\|_{[0,\alpha]}$$

by $\epsilon$'s definition. Now we have obtained

$$\|f^\star + r\|_{[0,\alpha]} < \|f^\star\|_{[0,\alpha]} \tag{3}$$

Now we turn to bounding the denominator of the objective function using $f^\star$ from above. Note that $\mathrm{sgn}\big(f^\star|_{[\alpha,1]}\big) = 1$ as $f^\star(\alpha) > 0$ and $f^\star$ has no roots in $[\alpha, 1]$ by Lemma 2. We will now show that in either case,

$$\min_{\lambda \in [\beta,1]}|(f^\star + r)(\lambda)| \geq \min_{\lambda \in [\beta,1]}|f^\star(\lambda)| \tag{4}$$

Case A: Since $\mathrm{sgn}(f^\star|_{M_k}) = 1$, $\mathrm{sgn}(r|_{M_k}) = -1$ which gives $\mathrm{sgn}\Big(r|_{[\lambda_{i_k},\beta]}\Big) = -1$ as $r$ has no roots in $(\lambda_{i_k}, \beta)$. Since $r$ has a simple root at $\lambda = \beta$ with no other roots greater than $\lambda = \beta$, we have that $\mathrm{sgn}\big(r|_{[\beta,1]}\big) = 1$. It immediately follows that

$$\min_{\lambda \in [\beta,1]}|(f^\star + r)(\lambda)| \geq \min_{\lambda \in [\beta,1]}|f^\star(\lambda)|$$

Case B: Since $\mathrm{sgn}(f^\star|_{M_k}) = -1$, $\mathrm{sgn}(r|_{M_k}) = 1$ which gives $\mathrm{sgn}\Big(r|_{[\lambda_{i_k},1]}\Big) = 1$ as $r$ has no roots in $(\lambda_{i_k}, 1)$. So $\mathrm{sgn}\big(r|_{[\beta,1]}\big) = 1$ and it follows that

$$\min_{\lambda \in [\beta,1]}|(f^\star + r)(\lambda)| \geq \min_{\lambda \in [\beta,1]}|f^\star(\lambda)|$$

Combining eq. (3) and eq. (4) gives

$$\frac{\|f^\star\|_{[0,\alpha]}}{\min_{\lambda \in [\beta,1]}|f^\star(\lambda)|} \geq \frac{\|f^\star + r\|_{[0,\alpha]}}{\min_{\lambda \in [\beta,1]}|(f^\star + r)(\lambda)|}$$

We note that $(f^\star + r)(0) = 0$ and $f^\star + r$ is an at most $t$-degree polynomial and therefore $f^\star + r$ is a feasible function for the minimization problem eq. (2). Thus, $f^\star$ is not a solution to eq. (2) as $f^\star + r$ is feasible and achieves a strictly smaller value for the objective function, which proves the lemma. □

**Lemma 4.** Define $\mathcal{P}''_t := \{f_t \in \mathcal{P}_t \mid f_t(0) = 0, f_t \text{ is } t\text{-equioscillatory on } [0,\alpha], f_t(\beta) = 1\}$ for $\beta > \alpha$. The problem

$$\underset{f_t \in \mathcal{P}''_t}{\text{minimize}} \max_{\lambda \in [0,\alpha]}|f_t(\lambda)|$$

is solved by

$$f^\star_t(\lambda) = \prod_{s=0}^{t-1} \frac{\lambda - r_{s,t}}{\beta - r_{s,t}}$$

$$r_{s,t} := \alpha \frac{\cos\left(\frac{\pi(s+1/2)}{t}\right) + \cos\left(\frac{\pi}{2t}\right)}{1 + \cos\left(\frac{\pi}{2t}\right)}$$

*Proof.* First note that $f_t^\star$ is indeed feasible; from inspection one realizes $f_t^\star(\beta) = 1$ and $r_{t-1,t} = 0$ (consequently $f_t^\star(0) = 0$) and as a variant of a Chebyshev polynomial (of the first kind), $f_t^\star$ is $t$-equioscillatory on $[0, \alpha]$ with extremal points

$$\gamma_s = \alpha \frac{\cos\left(\frac{\pi s}{t}\right) + \cos\left(\frac{\pi}{2t}\right)}{1 + \cos\left(\frac{\pi}{2t}\right)}, \quad 0 \le s \le t - 1$$

Assume for the sake of contradiction that there exists some $g \in \mathcal{P}_t''$ such that $\max_{\lambda \in [0,\alpha]} |g(\lambda)| < \max_{\lambda \in [0,\alpha]} |f_t^\star(\lambda)|$. This would then imply that the difference polynomial $h(\lambda) := f_t^\star(\lambda) - g(\lambda)$ satisfies the following

$$\forall s = 0, 2, \ldots : \quad h(\gamma_s) = f_t^\star(\gamma_s) - g(\gamma_s)$$
$$> f_t^\star(\gamma_s) - \max_{\lambda \in [0,\alpha]} |f_t^\star(\lambda)|$$
$$= 0$$
$$\forall s = 1, 3, \ldots : \quad h(\gamma_s) = f_t^\star(\gamma_s) - g(\gamma_s)$$
$$< f_t^\star(\gamma_s) + \max_{\lambda \in [0,\alpha]} |f_t^\star(\lambda)|$$
$$= 0$$
$$h(0) = f_t^\star(0) - g(0)$$
$$= 0$$
$$h(\beta) = f_t^\star(\beta) - g(\beta)$$
$$= 0$$

Since $h$ changes sign on every $\gamma_s$, it must have at least one root in each of the $t-1$ intervals between consecutive $\gamma_s$. By construction, $h$ also has a 2 more roots at 0 and $\beta$, meaning $h$ has at minimum $t + 1$ distinct roots. However, $h$ is a $t$th order polynomial, leading to a contradiction. □

### B.1 PROOF OF THEOREM 1

Proof. First, by Lemma 3, since any minimizer $f^\star$ of eq. (2) is $t$-equiosciallatory, all $t$ of $f^\star$'s roots lie in $[0, \alpha]$. This implies that $f^\star$ is increasing in $[\beta, 1]$ and therefore we have

$$\min_{\lambda \in [\beta,1]} |f^\star(\lambda)| = f^\star(\beta)$$

which is assumed to be positive without loss of generality also as in Lemma 3. Now we can scale $f^\star$ so that $f(\beta) = 1$ without changing the value of the objective function, i.e

$$\frac{\max_{\lambda \in [0,\alpha]} |f^\star(\lambda)|}{f^\star(\beta)} = \frac{\max_{\lambda \in [0,\alpha]} |Cf^\star(\lambda)|}{Cf^\star(\beta)} = \max_{\lambda \in [0,\alpha]} |Cf^\star(\lambda)|$$

where $C = \frac{1}{f^\star(\beta)}$. We have now transformed the problem into

$$\underset{f_t \in \mathcal{P}_t''}{\text{minimize}} \max_{\lambda \in [0,\alpha]} |f_t(\lambda)|$$

as in Lemma 4 which is solved by

$$f_t^\star(\lambda) = \prod_{s=0}^{t-1} \frac{\lambda - r_{s,t}}{\beta - r_{s,t}}$$

$$r_{s,t} := \alpha \frac{\cos\left(\frac{\pi(s+1/2)}{t}\right) + \cos\left(\frac{\pi}{2t}\right)}{1 + \cos\left(\frac{\pi}{2t}\right)}$$

as desired.

### B.2 SIMPLIFICATION OF EQUATION (1)

We begin by simplifying the original optimization problem

$$[\bar{U}] = \underset{[U] \in \text{Gr}(N,K)}{\text{argmin}} \left\| \bar{P} - UU^\top \right\|_F^2$$

$$
= \operatorname*{argmin}_{[\boldsymbol{U}] \in \mathrm{Gr}(N,K)} -2\left\langle \bar{\boldsymbol{P}}, \boldsymbol{U}\boldsymbol{U}^{\mathsf{T}} \right\rangle + \left\| \bar{\boldsymbol{P}} \right\|_{\mathrm{F}}^{2} + \left\| \boldsymbol{U}\boldsymbol{U}^{\mathsf{T}} \right\|_{\mathrm{F}}^{2}
$$

$$
= \operatorname*{argmin}_{[\boldsymbol{U}] \in \mathrm{Gr}(N,K)} -\left\langle \bar{\boldsymbol{P}}, \boldsymbol{U}\boldsymbol{U}^{\mathsf{T}} \right\rangle
$$

$$
= \operatorname*{argmin}_{[\boldsymbol{U}] \in \mathrm{Gr}(N,K)} -\left\langle \tilde{\boldsymbol{V}}\tilde{\boldsymbol{\Lambda}}\tilde{\boldsymbol{V}}^{\mathsf{T}}, \boldsymbol{U}\boldsymbol{U}^{\mathsf{T}} \right\rangle
$$

$$
= \operatorname*{argmin}_{[\boldsymbol{U}] \in \mathrm{Gr}(N,K)} -\left\langle \tilde{\boldsymbol{\Lambda}}, \tilde{\boldsymbol{V}}^{\mathsf{T}}\boldsymbol{U}\boldsymbol{U}^{\mathsf{T}}\tilde{\boldsymbol{V}} \right\rangle
$$

$$
= \left[ \tilde{\boldsymbol{V}} \operatorname*{argmin}_{\boldsymbol{W} \in \mathrm{St}(N,K)} -\left\langle \tilde{\boldsymbol{\Lambda}}, \boldsymbol{W}\boldsymbol{W}^{\mathsf{T}} \right\rangle \right]
$$

$$
= \left[ \begin{bmatrix} \boldsymbol{V} & \boldsymbol{V}_{\perp} \end{bmatrix} \operatorname*{argmin}_{\boldsymbol{W} \in \mathrm{St}(N,K)} -\left\langle \begin{bmatrix} \boldsymbol{\Lambda} & \boldsymbol{0} \\ \boldsymbol{0} & \boldsymbol{\Lambda}_{\perp} \end{bmatrix}, \boldsymbol{W}\boldsymbol{W}^{\mathsf{T}} \right\rangle \right]
$$

Since $\boldsymbol{W} \in \mathrm{St}(N,K)$, it follows that $\mathrm{tr}\left( \boldsymbol{W}\boldsymbol{W}^{\mathsf{T}} \right) = K$ and all diagonal elements of $\boldsymbol{W}\boldsymbol{W}^{\mathsf{T}}$ lay in the range $[-1,1]$. Consequently, $\min_{\boldsymbol{W} \in \mathrm{St}(N,K)} -\left\langle \tilde{\boldsymbol{\Lambda}}, \boldsymbol{W}\boldsymbol{W}^{\mathsf{T}} \right\rangle \geq -\mathrm{tr}(\boldsymbol{\Lambda})$. Since $\boldsymbol{W} = \begin{bmatrix} \boldsymbol{Q} \\ \boldsymbol{0} \end{bmatrix}$ for arbitrary $\boldsymbol{Q} \in \mathrm{St}(K,K)$ satisfies this inequality with equality, it may be substituted as the solution and a final simplification completes the proof.

$$
\begin{aligned}
[\bar{\boldsymbol{U}}] &= \left[ \begin{bmatrix} \boldsymbol{V} & \boldsymbol{V}_{\perp} \end{bmatrix} \operatorname*{argmin}_{\boldsymbol{W} \in \mathrm{St}(N,K)} -\left\langle \begin{bmatrix} \boldsymbol{\Lambda} & \boldsymbol{0} \\ \boldsymbol{0} & \boldsymbol{\Lambda}_{\perp} \end{bmatrix}, \boldsymbol{W}\boldsymbol{W}^{\mathsf{T}} \right\rangle \right] \\
&= \left[ \begin{bmatrix} \boldsymbol{V} & \boldsymbol{V}_{\perp} \end{bmatrix} \begin{bmatrix} \boldsymbol{Q} \\ \boldsymbol{0} \end{bmatrix} \right] \\
&= [\boldsymbol{V}\boldsymbol{Q}] \\
&= [\boldsymbol{V}]
\end{aligned}
$$

