# OpenReview forum: "Rapid Grassmannian Averaging with Chebyshev Polynomials"
_ICLR.cc/2025/Conference — Submitted to ICLR 2025_

### Official Review · Reviewer_192F · 2024-10-28

**Soundness:** 4
**Presentation:** 3
**Contribution:** 2
**Rating:** 3
**Confidence:** 4

**Summary:**

The paper provides a fast algorithm for computing a particular average on the Grassmann manifold. The algorithm is, in essence, a variant of the classic power iteration, which is tailored for projection matrices (which can, in turn, be used to represent points on the Grassmannian). The algorithm is suitable for decentralized computations, making it easy to scale across machines. Empirical results show that the algorithm is faster than a selection of baseline algorithms while providing comparable results.

**Strengths:**

The paper observes that the eigenvalues of projection matrices have a particular structure, which can be exploited to build a more efficient variant of the power iteration to compute eigenvalues/vectors. As far as I can tell, this contribution is novel and is potentially more widely applicable than explored in the submitted paper.

The paper is generally well-written and easy to understand.

The developed algorithm is shown to be easy to extend to a decentralized implementation. This is a quite nice practical property.

**Weaknesses:**

My main concern with the paper is the studied problem. The paper states that points on the Grassmann manifold are "used ubiquitously in machine learning, computer vision, and signal processing". It is true that some years ago (before deep learning took off), some work went into using subspaces (and hence Grassmannians) to represent batches of data. This approach has, however, not caught on and is quite rare today.

I do not want to argue that a paper is uninteresting because it explores a (current) niche approach. However, I consider it problematic that the paper does not demonstrate that the problem of study is relevant to solve. The paper demonstrates that the proposed algorithm is faster than alternatives, but it fails to demonstrate that the algorithm computes a quantity of interest. I would consider the paper significantly more interesting if it was demonstrated that Grassmann Averaging is useful for solving real tasks.

Some further minor comments:
* It seems to me that Theorem 1 is more generally applicable than just computing Grassmann averages. If this understanding is correct, it would have been valuable to have pointers in such directions.
* I miss references to early work on using Grassmann Averages to define principal components of data: "Grassmann Averages for Scalable Robust PCA", CVPR 2014. This relies on a crude chordal-like metric.
* While I appreciate that many algorithms are provided in the paper, I miss general explanations of what these algorithms do. On a first reading, it can be difficult to read an algorithm, so I would have appreciated more hand-holding.

**Questions:**

1. Can you clarify what is meant by "QR" in the first equation of Sec. 3.3? I know the QR decomposition, but this generally returns *two* values (the Q and the R matrices). In the text, only a single matrix is returned, so it would be good to clarify if this was Q or R.
2. Can you provide a citation for "StableQR"?
3. In the last experiment, you compute a Frechet mean, but use the chordal distance. Isn't the Frechet mean defined according to the geodesic distance? Can you clarify what is actually being done here?

---

> ### Author Response · Authors · 2024-11-27
> **Overall Response**
>
> Thank you for your review. We will try to better motivate the paper with more examples of practical applications. Here is a brief overview of some applications we intend to include and elaborate upon:
> * Covariance Estimation in Array Processing
> * Principal Component Analysis
> * Low-Rank Matrix Completion
> * Distance/Metric Learning
> * Multi-Task Feature Learning
> * Subspace Clustering
>
> Regarding the "Grassmann Averages for Scalable Robust PCA" paper, we will incorporate this into our Related Work section. We will also attempt to improve the “hand-holding” when detailing our algorithm.
>
> To address your questions individually,
>
> ___
> * *Can you clarify what is meant by "QR" in the first equation of Sec. 3.3? ...*
>
> The function returns the Q matrix from a QR factorization. We assumed the notation would be clear from the dimension of the returned quantity and context, as we have seen similar notation used in the literature, however we will add an explicit description to the paper to avoid any confusion.
>
> ___
> * *Can you provide a citation for "StableQR"?*
>
> The StableQR algorithm was a simple fix we came up with to mitigate the pitfalls of standard QR implementations, there is no citation.
>
> ___
> * *In the last experiment, you compute a Frechet mean ...*
>
> The chordal distance is only used for cluster assignment (each subspace is assigned to its closest cluster center w.r.t. the Grassmannian chordal distance). The Frechet mean uses Grassmannian logarithm/exponent maps as per usual.
>
> We can change this to just use the normal metric since that simplifies the explanation. I used chordal since it does nearly the same thing and there are a lot of points to assign to cluster, so for prototyping I wanted it to be quick.

---

> > ### Comment · Reviewer_192F · 2024-11-27
> > **Thanks**
> >
> > I appreciate the clarifications. As to the last point, I agree that the chordal metric isn't too dissimilar from the geodesic one, so I understand the practical choice. I would, however, value a consistent choice of metric.
> >
> > Regarding the motivation then I think these are great improvements. However, my request was a bit more than motivation. I think the paper would be notably stronger if you could **demonstrate** an empirical improvement over *some* state-of-the-art. I want to emphasize that I do not concern myself too much with the choice of application or the associated state-of-the-art, but I would like to see some notion of "we could not do this before".

---

### Official Review · Reviewer_bp4e · 2024-11-02

**Soundness:** 2
**Presentation:** 2
**Contribution:** 2
**Rating:** 3
**Confidence:** 3

**Summary:**

The paper proposes a method for computing the average of some subspaces in the standard and decentralized setting.

**Strengths:**

omitted.

**Weaknesses:**

- The contribution at the technical level is incremental, in my opinion. Theorem 1 seems to be a basic variant of the known result for Chebyshev polynomials. The only difference seems to be that the paper adds an extra constraint $f_t(0)=0$.

- The assumption on the dual-banded spectrum is also not new. This assumption is used in the AISTATS 2022 paper: Super-Acceleration with Cyclical Step-sizes (https://proceedings.mlr.press/v151/goujaud22a/goujaud22a.pdf); see Eq. 3 there. That paper has a much deeper exploration of related aspects than the present manuscript. If NeurIPS holds the same high standard, the lack of technical depth forms a ground for rejection.

- From a practical viewpoint, the proposed method performs similarly to the baselines it compares, and there is no significant advantage in terms of acceleration or efficiency, as shown in Section 5.1

The above two points suggest Lines 60 - 62 are never-before-seen overstatements: "*Our algorithms operate similarly to the famous power method, with the distinction that Chebyshev polynomials are employed to leverage a “dual-banded” property of the problem in order to achieve never-before-seen efficiency in computation and communication.*"

- The algorithm listings (Algorithms 1 and 2) are highly repeated.
- Section 5.2 is disconnected from the prior sections of the paper, which puts emphasis on the decentralized setting. From Section 1 to Section 5.1 there is no mention of k-means clustering.

**Questions:**

omitted.

---

> ### Author Response · Authors · 2024-11-27
> **Overall Response**
>
> Thank you for your review. To address your points individually,
>
> ___
> * *The contribution at the technical level is incremental, in my opinion...*
>
> Chebyshev polynomials indeed have many known special properties causing them to appear often in mathematics, however we believe the assessment of Theorem 1 as “basic” may be an underestimation. To the best of our knowledge, this result has not been previously established. Our primary contributions are proving the independence of solution to upper threshold $\beta$, proving under the constraint of a fixed root, and the recognition that the modified variants are well-approximated for arbitrary order via a recursive process. The proof of Theorem 1 allows us to confidently claim that our algorithm will outperform any other polynomial-based algorithm (e.g. DeEPCA) provided $\alpha$ is chosen appropriately.
>
> While specific problems such as decentralized PCA have been studied thoroughly in the past, our paper provides an efficient algorithm for the generalized problem of averaging points on a Grassmannian. Algorithms of this generalized form have been studied less, and as such we believe there is merit to providing a more versatile, general solution.
>
> ___
> * *The assumption on the dual-banded spectrum is also not new...*
>
> We agree that the concept of dual-banded spectra is not new, nor do we intend to claim it as so; we hope this did not come across improperly in our paper. If there was a specific region or statement in our paper that improperly conveyed this message, please point it out to us so we may correct it. What we intend to claim in our paper is that the problem of Grassmannian averaging also involves such dual-banded matrices.
>
> Thank you for your reference to the AISTATS paper. While said paper also mentions dual-banded matrices, it is considering them from a very different perspective, that of the structure of Hessian matrices in the context of deep learning problems. Critically, their dual-banded consideration is much less general than ours (e.g. they require the bands be of equal size, we make no such restriction) and focuses on relevance to step sizes in iterative descent algorithms. The dual-banded assumption is not as critical to our analysis; our Theorem 1 establishes that indeed only the size $\alpha$ of the lower band impacts our algorithm’s parameters. For these reasons, we do not believe this should be viewed as a shortcoming of our paper.
>
> ___
> * *From a practical viewpoint, the proposed method performs ...*
>
> In Section 5.1, we indeed show that our algorithm provides a significant advantage. For instance, Figure 2(a) shows our method beating all other baselines by orders of magnitude after 30 rounds of communication. Table 1 shows our method solves problems in less time than all other baselines for all but one numerical tolerance. If your claim is that the convergence rate of our algorithm is not better than that of the most competitive baseline, DeEPCA, (both appear to converge linearly) we believe this ignores the practical consideration of simple runtime; a faster algorithm is preferable to a slower one, all else being equal.
>
> ___
> * *The algorithm listings (Algorithms 1 and 2) are highly repeated.*
>
> We will consider delegating one to the appendix in the revision.
>
> ___
> * *Section 5.2 is disconnected from ...*
>
> The experiments of Section 5.2 were intended to present results of our algorithm applied to real world data. Our algorithm is intended to be a general solution to the problem of averaging points on a Grassmannian, one which may be used as an intermediate step to solve higher-level problems such as clustering on a dataset. We will try to motivate this better in the revision.

---

### Official Review · Reviewer_J9cC · 2024-11-03

**Soundness:** 3
**Presentation:** 3
**Contribution:** 2
**Rating:** 5
**Confidence:** 3

**Summary:**

This paper addresses the problem of computing the induced arithmetic mean of subspaces (points on the Grassmannian manifold). The authors focus on a decentralized setting where each agent has limited data and restricted communication with its neighbors. Their approach is based on spectral analysis of the associated optimization problem, resulting in a power-method-like algorithm. The utility of the method is demonstrated through synthetic experiments and an application to clustering video motion data via Grassmannian K-means clustering.

**Strengths:**

- The paper addresses an interesting problem and is generally well-written, presenting the proposed approach clearly, though it occasionally lacks detail.
- The authors prove that Chebyshev polynomials are optimal for their modified power method.
- Their experiments demonstrate that the algorithm outperforms competing methods in synthetic Grassmannian averaging and K-means clustering of subspaces spanned by video sequences for motion clustering. They have also successfully adapted related methods (such as DeEPCA) to the problem of Grassmannian averaging, which can be seen as part of their contribution.
- The authors provide code that implements the proposed approach and reproduces the experiments presented in the paper (although I have not tested the code).

**Weaknesses:**

1. Importance of Decentralized Grassmannian Averaging: The paper emphasizes the importance of decentralized averaging, which the authors claim is significant. However, the paper does not provide concrete examples where decentralization is essential and demonstrates it only through a single set of synthetic experiments (emulating a decentralized setup). Could you provide more compelling examples to illustrate the importance of this problem?

2. Comparison and Relation to Existing Literature: How does RGrAv compare or relate to existing methods, such as block power methods (e.g., Chebyshev-Davidson, spectral filtering) or other similar approaches? Given that the problem in Eq. (1) is reduced to a spectral problem in Section 3.1, a more thorough exploration of relevant methods, both centralized and decentralized, would be beneficial. (I am not an expert in this area, but there appears to be relevant literature available.)

3. Need for Additional Explanations: While the paper is overall well-written, certain choices or statements would benefit from further explanation or justification. I was able to understand some of these upon reflection, while others remain unclear. For example:
  - Section 3.1: The definition of the optimization problem (1) and the equivalent spectral perspective are presented very densley, without references, which could be difficult for readers unfamiliar with the topic. For instance, why is (1) invariant to the choice of representatives $[U_m]$? Why is the projection on the K largest eigenvectors $[V]$ the minimizer of (1)? Additional explanations or citations would be helpful.
  - Section 3.2: What is average consensus (AC)? Without prior knowledge, the proposed procedure is unclear. Why can’t AC be used directly to approximate (1)? Most importantly, why is it sufficient to approximate the lower dimensional $PX$? In what way does this guarantee an approximation of $P$?
  - Section 4.1: How are the optimal polynomials $f_t$ used to recover the space $U$? This is again central to the approach, but is challenging to understand. Theorem 1 proves the optimality of $f_t$ in the sense of Eq. (2), but in what sense does this translate to the optimality of the approximate average space $[U]$?
  - Section 4.2: "fortunately, $f_t$ is well-approximated in terms of $f_{t-1}^*$ and $f_{t-2}^*$, in what sense does this hold? Is it an experimental observation or is there a theoretical justification?

4. Experimental Results Are Limited:
  - The experiments in Section 5.1 focus on the decentralized setting and raise a few questions:
    - In these experiments, DRGrAv is "sub-optimally tuned" (Line 353) while parameters for other methods are carefully tuned through parameter search (commendable effort). How would DRGrAv perform if optimally tuned?
    - The experiments are conducted for a single setup (M,N,K)=(64,150,30) with two types of agent connectivity (hypercube and cycle). Is this setup representative?
    - Are similar results obtained against competitive methods in different setups?
    - Is this a setup where decentralization is necessary or preferable? How would DRGrAv perform compared to competitors in a scenario with very large M or N, where decentralization is crucial?
    - Additional experiments supporting the importance and benefits of the proposed approach for decentralized averaging would significantly strengthen the manuscript, given that this is a primary focus of the paper.

  - The experiment in Section 5.2 uses Grassmannian averaging to define Grassmannian K-means for motion clustering in videos. The authors present only timing results but do not include any assessment of clustering quality for each method. The statement that "the four averaging algorithms produce clusters with similar quality across various values for K  (these results are not shown for brevity)" is unclear. Showing these results alongside runtime comparisons seems essential for evaluating each method's overall effectiveness. These results could be briefly mentioned in the main text and detailed in the appendix if space is limited. Does RGrAv produce the same results as Flag? How does RGrAv compare to the block power method, ground truth IAM? Does the spectrum gap assumption (Section 4.1) hold in these experiments?

  - The evidence for a performance advantage is somewhat limited. The decentralized results in Section 5.1 are comparable to DeEPCA, while the centralized results in Section 5.2 are closely followed by the block power method.

**Questions:**

Additional issues and questions:
- Eq. 1: It would be good to add parantheses to clarify whether summation includes $UU^T$ to avoid potential confusion.
- Could you elaborate on the connection between Theorem 1 and the equioscillation theorem? Would this reproduce the best polynomial approximation of the step function?
- Please explain how the DeEPCA algorithm is adapted for decentralized Grassmannian averaging.
- Line 172: Unrolling the power method loop—could you clarify if this is straightforward or requires further justification?
- Line 198: Are you assuming that $\alpha$ and $\beta$ are known?
- Figure 1: Please elaborate on what is shown and its significance. Additionally, Figure 1 is not referenced in the main text.
- Line 264: "focuses on minimizing the error in the gradient tracking procedure" — please provide more details.
- Table 1: Are these results obtained under the hypercube or cycle setup?
- Line 438: How is the true IAM average computed? Could you include timing for this computation as well?
- Section 5.2: Please clarify that this experiment does not test the decentralized algorithm.
- $\alpha$ is used as a bound on eigenvalues in Section 4 and later as a step size parameter in Section 5, which could be confusing. Please consider revising this notation.

Additional minor issues:
- Line 51: "subsequent projections may be applied locally thereafter)." - could you please elaborate?
- Line 62: "We demonstrate merit through a theoretical guarantee on the optimality" - Theorem 1 proves optimality of the choice for the polynomial $f_t$. Does this guarantee the optimality of your approach? In what sense?
- Line 72: Define Grassmannian and Stiefel manifolds, as these terms and notations are introduced here for the first time.
- line 377: What is the parameter $\rho$?

---

> ### Author Response · Authors · 2024-11-27
> **Overall Response [Part 1]**
>
> [Part 1]
>
> Thank you for your review. To address your points individually,
>
> ___
> * *Importance of Decentralized Grassmannian Averaging: ...*
>
> We will attempt to better motivate situations where decentralization is essential. Briefly, these are situations when the data is sufficiently large, must respect some privacy model, or the algorithm is run on edge-devices. Around line 53 we attempted to detail how operating on sufficiently large data may necessitate the use of distributed algorithms, but we agree this could be elaborated upon. Additionally, we intend to include further concrete applications; these include covariance estimation in array processing, PCA, low-rank matrix completion, distance/metric learning, multi-task feature learning, and subspace clustering.
>
> ___
> * *Comparison and Relation to Existing Literature: ...*
>
> We will include these considerations in the revision.To briefly address the question, methods such as Chebyshev-Davidson tend to be hard to decentralize due to intermediate computations of quantities such as solutions to inverse problems and eigenproblems. In the specific case of Chebyshev-Davidson, the solved problem is actually semantically different as well: extract specific non-leading eigenpairs. Since our algorithm, from a spectral perspective, seeks only to learn only the span of the leading eigenvectors, we can compute this more efficiently.
>
> ___
> * *Section 3.1: ...*
>
> We will address and elaborate upon this in the revision. To address these questions briefly,
> * The invariance of choice of representative comes from the fact that $(U_m Q) (U_m Q)^T = U_m U_m^T$ for any square orthogonal matrix $Q$.
> * The eigenvector solution comes from the fact that problem (1) is essentially a type of Procrustes problem (see “Procrustes Problems” by John Gower and Garmt B Dijksterhuis). A proof will be given in the revision.
>
> ___
> * *Section 3.2: ...*
>
> To address these concerns:
> * *What is average consensus (AC)?* -
> Average consensus is used frequently in the literature of decentralized algorithms, so we thought introducing it with reference as we did in line 137 was sufficient: “Average consensus (AC) is a useful primitive in decentralized optimization to quickly approximate the average of real numbers in a decentralized manner (Nedic & Ozdaglar, 2009).”
> * *Why can’t AC be used directly to approximate (1)?* -
> Line 139 states “Unfortunately, the non-convex manifold structure of Gr(N, K) precludes us from efficiently applying AC directly to solve eq. (1)”; if a set is not convex, then it is a fact that the average of points from this set may not necessarily be a member of the set. So AC would not compute an average on the manifold.
> * *Most importantly, why is it sufficient to approximate the lower dimensional $PX$? In what way does this guarantee an approximation of $P$?* -
> This is elaborated upon later in Section 3.3, as power methods only require matrix-vector products to function. We will try to better motivate this in Section 3.2 however to prevent prematurely confusing the reader.
> We will attempt to clarify these points in the revision.
>
> ___
> * *Section 4.1: ...*
>
> We will attempt to clarify this in the revision. Briefly, our method is akin to noise cancellation. We start with a matrix like $\bar{P}$, then perturb its eigenvalues to try to “cancel” out the trailing eigenvalues relative to the leading eigenvalues. Any collection of $K$ vectors applied to such a matrix would then approximately lie in the span of the leading eigenvectors $[V]$. Since polynomials applied to matrices preserve eigenvectors while applying the polynomial to each eigenvalue, identifying an ideal polynomial in the sense of Theorem 1 leads to convergence on $[U]$.
>
> ___
> * *Section 4.2: ...*
>
> It is an experimental observation, partially justified visually with Figure 1. We will make this clear in the revision.
>
> ___
> * *Experimental Results Are Limited: …*
>
> To address these points succinctly, we will rework things to try to address all criticisms mentioned. To directly address one point: if optimally tuned, our method DRGrAv will converge even quicker. However, we believe being able to optimally tune DRGrAv is not realistic in practice, hence the use of a realistic heuristic tuning. Baseline methods were optimally tuned to ensure that our performance was not merely a consequence of poor tuning of the baselines. In practice, one would have to heuristically tune these as well; in some sense, this makes the performance of our algorithm look artificially poorer by comparison in these experiments. However, we believe this approach should give readers confidence the comparison is not “cherry-picked”.

---

> ### Author Response · Authors · 2024-11-27
> **Overall Response [Part 2]**
>
> [Part 2]
>
> ___
> * *Eq. 1: ...*
>
> We can do this.
>
> ___
> * *Could you elaborate on the connection between Theorem 1...*
>
> Could you provide a more specific question? Equioscillation is considered in the appendix, mainly Definition 1, Lemma 3, and Lemma 4. Whether or not the solution presented in Theorem 1 is the best polynomial approximation to the step function is an interesting quandary, and one we pondered in the process of writing this paper. Essentially, we have to first define what we mean by “best,” for which we can imagine several distinct reasonable criteria. If by “best” we mean minimizing the ratio of equation (2), then the answer is yes and this is the result of Theorem 1. We may include some additional language in the revision to this effect.
>
> ___
> * *Please explain how the DeEPCA ...*
>
> This will be addressed in the revision. In short, we use the fixed-size bases $U_m$ as the data matrices input to the DeEPCA algorithm. We also replace the SignAdjust procedure with StableQR.
>
> ___
> * *Line 172: ...*
>
> We can include language to explain why the QR factorization need only be applied at the end. Aside from this, we believe line 172 should follow fairly straightforwardly from line 168; please let us know if there are other points of confusion we should address here.
>
> ___
> * *Line 198: ...*
>
> We will try to be more explicit here that the values exist, but are estimated heuristically in practice.
>
> ___
> * *Figure 1: ...*
>
> This will be addressed in the revision.
>
> ___
> * *Line 264: ...*
>
> This will be addressed in the revision.
>
> ___
> * *Table 1: ...*
>
> Hypercube; this will be addressed in the revision.
>
> ___
> * *Line 438: ...*
>
> The true IAM is computed by computing the leading eigenvectors of $\bar{P}$ directly using the standard torch.linalg.eigh function in python. We will include timing for this operation in the revision.
>
> ___
> * *Section 5.2: ...*
>
> It does not, it indeed tests the centralized variant.
>
> ___
> * *$\alpha$ is used ...*
>
> This choice was made to align with the notation of the papers cited in Section 5. We will fix this in the revision.
>
> ___
> * *Line 51: ...*
>
> This will be addressed in the revision. In short, computing the IAM can be thought of as a two-step process: first compute a Euclidean average, second project back onto the manifold. This sentence was just addressing the fact that in a decentralized setting, once all agents have computed the Euclidean average, the projection operation may be performed locally without further inter-agent communication.
>
> ___
> * *Line 62: ...*
>
> We will try to make this more clear in the revision. In short, in the space of iterative power method-like polynomial algorithms, yes. If one chooses $\alpha$ appropriately, then our algorithm will converge faster than any other such algorithm, e.g. DeEPCA.
>
> ___
> * *Line 72: ...*
>
> This will be addressed in the revision.
>
> ___
> * *line 377: ...*
>
> It is a parameter that represents a regularization term penalty from the “A riemannian gossip approach to subspace learning on grassmann manifold” paper by Mishra et. al. We will try to clarify that these parameters are using the namespace of the paper from which the algorithm is defined.

---

> > ### Comment · Reviewer_J9cC · 2024-11-27
> >
> > Thank you for the detailed responses!
> >
> > When are you planning to upload the revised manuscript? Based on your answers you are planning significant edits, which I look forward to reading, but may be challenging to complete by the revision deadline (tomorrow).

---

### Meta-Review · Area_Chair_ajUd · 2024-12-22

**Metareview:**

This paper introduces an efficient approach for performing subspace averaging on the Grassmannian manifold in a decentralized setting. The authors provide theoretical guarantees of optimality and empirical results showcasing the proposed algorithm’s advantages in terms of accuracy and runtime compared to relevant approaches. Although the paper is well-organized and easy to follow, the authors have not sufficiently emphasized the significance of their contributions. The reviewers raised concerns regarding the practical relevance of the proposed method, the novelty of its technical ideas, as well as the thoroughness of its experimental validation and comparisons with existing solutions. In my view, the paper requires major revisions before being ready for acceptance at ICLR. Specifically, the authors should more clearly motivate their approach, emphasize the importance of the problem, improve the clarity of the technical contributions, and strengthen the empirical validation section. That being said, I recommend rejecting this paper in its current form.

**Additional Comments On Reviewer Discussion:**

During the rebuttal phase, the authors responded to reviewer J9cC’s concerns regarding the importance of the problem, the presentation of technical ideas, comparisons with similar existing methods, and details in the experimental section. While they promised to revise the paper accordingly, they did not update the manuscript before the deadline. Additionally, the authors addressed the comments of reviewers bp4e and 192F regarding the significance and relevance of the work, but they failed to convince the reviewers to increase their scores. Overall, the rebuttal phase demonstrated that the paper could be significantly improved by addressing the aforementioned issues; however, these revisions would require major changes to the paper’s presentation. Therefore, the current version of the paper cannot be accepted.

---

### Decision · Program_Chairs · 2025-01-22

Reject